# Integration of molecular coarse-grained model into geometric representation learning framework for protein-protein complex property prediction

Yang Yue[1,6], Shu Li[2,6], Yihua Cheng[1], Lie Wang [3], Tingjun Hou [4], Zexuan Zhu [5] ✉ & Shan He[1,2] ✉

Structure-based machine learning algorithms have been utilized to predict the properties of protein-protein interaction (PPI) complexes, such as binding affinity, which is critical for understanding biological mechanisms and disease treatments. While most existing algorithms represent PPI complex graph structures at the atom-scale or residue-scale, these representations can be computationally expensive or may not sufficiently integrate finer chemical-plausible interaction details for improving predictions. Here, we introduce MCGLPPI, a geometric representation learning framework that combines graph neural networks (GNNs) with MARTINI molecular coarse-grained (CG) models to predict PPI overall properties accurately and efficiently. Extensive experiments on three types of downstream PPI property prediction tasks demonstrate that at the CG-scale, MCGLPPI achieves competitive performance compared with the counterparts at the atom- and residue-scale, but with only a third of computational resource consumption. Furthermore, CG-scale pre-training on protein domain-domain interaction structures enhances its predictive capabilities for PPI tasks. MCGLPPI offers an effective and efficient solution for PPI overall property predictions, serving as a promising tool for the large-scale analysis of biomolecular interactions.

Protein-protein interactions (PPIs) play a pivotal role in regulating diverse cellular processes, including signal transduction, immune response, and metabolic regulations[1,2]. Gaining insights into PPI aids in understanding protein functions and identifying potential drug targets[3–6]. While traditional experimental techniques for studying PPIs, such as yeast two-hybrid screening[7], co-immunoprecipitation[8], pull-down assays[9], and fluorescence resonance energy transfer (FRET)[10], are effective, they often require extensive labor and substantial financial investment. To address these challenges, advancements in computational tools and artificial intelligence (AI) algorithms have transformed the study of PPIs[11]. These in-silico strategies leverage expansive datasets to predict PPIs, enabling interaction site prediction[12], interaction type classification[9], and binding affinity prediction[13].

The three-dimensional (3D) structures of proteins are fundamental to their biological functions[14–16]. To gain a nuanced understanding of the biological significance and detailed mechanisms underlying PPIs, decoding the geometry of protein complexes have

[1]School of Computer Science, The University of Birmingham, Edgbaston, Birmingham, UK. [2]Macao Polytechnic University, Macao, China. [3]Bone Marrow Transplantation Center of the First Affiliated Hospital, Institute of Immunology, Zhejiang University School of Medicine, Hangzhou, China. [4]College of Pharmaceutical Sciences, Zhejiang University, Hangzhou, China. [5]National Engineering Laboratory for Big Data System Computing Technology, Shenzhen University, Shenzhen, China. [6]These authors contributed equally: Yang Yue, Shu Li. ✉e-mail: zhuzx@szu.edu.cn; s.he@cs.bham.ac.uk

become essential[1]. Among various computational methods, Graph neural networks (GNNs)[13,17] stand out with their proficiency in handling the 3D structures of proteins. By integrating spatial information and topological data inherent to protein complexes, GNNs provide a robust framework for illuminating the multifaceted nature of protein interactions[18,19]. For instance, Jing et al.[20] proposed a GNN framework, GVP-GNN, which preserves rotation equivariance of protein rigid motions when capturing geometric representations of protein-protein complexes. Zhang et al.[21] designed a line-graph-augmented message passing scheme to inject the relative positional information between two interactive edges for different PPI prediction tasks, such as protein-protein interface identifications.

Notably, in GNNs-based methods, proteins are represented as graph structures, with nodes corresponding to either heavy atoms (i.e., the atom-scale model) or amino acids (i.e., the residue-scale model)[21,22]. However, each approach has its own trade-offs. Atom-scale models, while detailed, demand extensive computational resources to manage thousands of nodes, limiting their application to large PPI systems. On the other hand, residue-scale models are more computationally tractable but may overlook critical binding details that influence specificity and affinity. To address these limitations, multi-scale information can be integrated into the node features and edge connections. However, such integration requires intricate information exchange across scales, maintaining model consistency and physical relevance, which can complicate the design process. Additionally, in both atom- and residue-scale models, edges typically represent interactions based on sequential threshold or geometric distance, aiming to capture the complex relationships between protein structures and functions. Nevertheless, using such criteria to define connections may misrepresent chemical bonds, potentially affecting predictive accuracy.

A potential solution to these issues is to adopt coarse-grained (CG) modeling, which is a well-established framework in protein molecular dynamics (MD) simulation, designed to effectively strike a balance between maintaining essential molecular details and enhancing computational efficiency. CG-scale representation simplifies groups of atoms into single sites, such as amino acid side chains or specific chemical groups. The MARTINI model[23,24], a widely recognized CG-scale model in protein MD simulation, represents an average of four heavy atoms and their associated hydrogens with a single CG bead. It classifies beads into multiple main physical types, including polar (P), nonpolar (N), apolar (C), and charged (Q), etc., with subtypes based on hydrogen bonding capabilities or polarity. In addition to the various bead types, the model includes numerous chemical-plausible interaction parameters, both bonded (bonds, angles and dihedrals) and nonbonded, to directly and accurately reflect the partitioning free energy of amino acid sidechains[24,25]. Through this strategy, the MARTINI model retains essential molecular interaction features while significantly reducing computational demands. It has been successfully applied and evaluated in many PPI-related studies, including the dimerization of the amino acid side chains[26], interactions involving membrane proteins such as glycophorin A[27], G protein-coupled receptor (GPCR) rhodopsin[28,29], and the Epidermal Growth Factor Receptor complex[30], as well as soluble protein complex interactions[29] like insulin, Ras-Raf, and Barnase-Barstar. Additionally, the MARTINI force field has been integrated into the HADDOCK framework[31], enhancing its capability to predict the 3D structures of protein interactions.

Although the CG-scale offers improved efficiency, its simulations still consume more resource than PPI predictions using AI techniques. Previous efforts to integrate the CG-scale model with machine learning (ML) or deep learning (DL) methods have primarily focused on optimizing force field potential parameters, predicting the peptide self-assembly shapes, and converting the CG-scale model back to atomistic structures[32,33]. However, a comprehensive approach that combines AI and CG modeling to predict PPI properties remains an under-explored area.

In this study, we present MCGLPPI, a lightweight geometric representation learning framework that combines GNNs with the MARTINI CG-scale models to predict the overall properties of PPI complexes. Designed to optimize computational efficiency without compromising prediction accuracy, MCGLPPI employs a specially-designed CG-scale complex graph, which maps each CG bead of protein complexes to nodes and utilizes chemical-plausible MARTINI force field bond parameters as edges for efficient structural characterization. Additionally, we introduce a GNN-based CG geometric-aware encoder to extract the high quality representations from the devised graph.

Our extensive validation demonstrates that MCGLPPI achieves competitive performance on multiple curated overall property prediction benchmarks for PPI structures, including binding affinity relevant prediction and interaction type classification tasks. When compared to atom-scale and residue-scale counterparts, MCGLPPI significantly improves computational efficiency, and reduces graphics processing unit (GPU) usage and total running time by more than threefold without compromising accuracy. Moreover, proteins are intricate molecular machines that typically consist of multiple domains. Domain-domain interactions (DDIs) are critical subsets of PPIs, where the interaction typically occurs between domains rather than the entire proteins[34–36]. We demonstrate that the CG-scale pre-training based on DDI patterns effectively enhances the model's ability to predict PPI binding affinities. Overall, MCGLPPI emerges as a general, accurate, and efficient method for predicting PPI properties, offering a pathway to sophisticated analysis of biomolecular interactions.

## Results

### Overview of the proposed CG-scale complex geometric learning framework

We integrate biomolecular CG structures, force field parameters, and geometric-aware GNNs within the MCGLPPI framework for efficient prediction of overall properties of protein-protein complexes, which consists of three major components: (1) CG-scale complex graph generation, (2) CG-scale geometric representation learning, and (3) DDI-based CG-scale graph encoder pre-training. A comprehensive overview of the framework and its components is provided in Fig. 1.

### Force field parameter and CG-scale complex graph generation

Structure-based prediction of PPI complex properties typically demands high-quality learning of protein geometric graph representations. The number of graph nodes and edges significantly affect computational cost. At the same time, it is crucial to ensure that the graph structure is chemically plausible, as it is essential for accurately depicting the properties of protein complexes.

On top of this, we introduce the CG scale-based MARTINI parameterization that aims to efficiently achieve a balanced representation between chemically-plausible interaction characterization and computational cost. This process commences by transforming an atomistic PPI structure into the CG-scale structure and a comprehensive set of CG-scale force field parameters tailored for the MARTINI model (the extensively-used MARTINI22[23,37] and the latest MARTINI3[24] are examined, a crucial difference is that MARTINI3 brings richer bead types and bead numbers to slightly increase the bead resolution, and their detailed version characteristics are in Supplementary Note 1). This simplification reduces the high-resolution atomic model into a computationally easier-to-execute form by grouping multiple atoms into fewer representative beads. The resulting parameters describe how these beads interact with each other chemically and physically from different perspectives (Fig. 2).

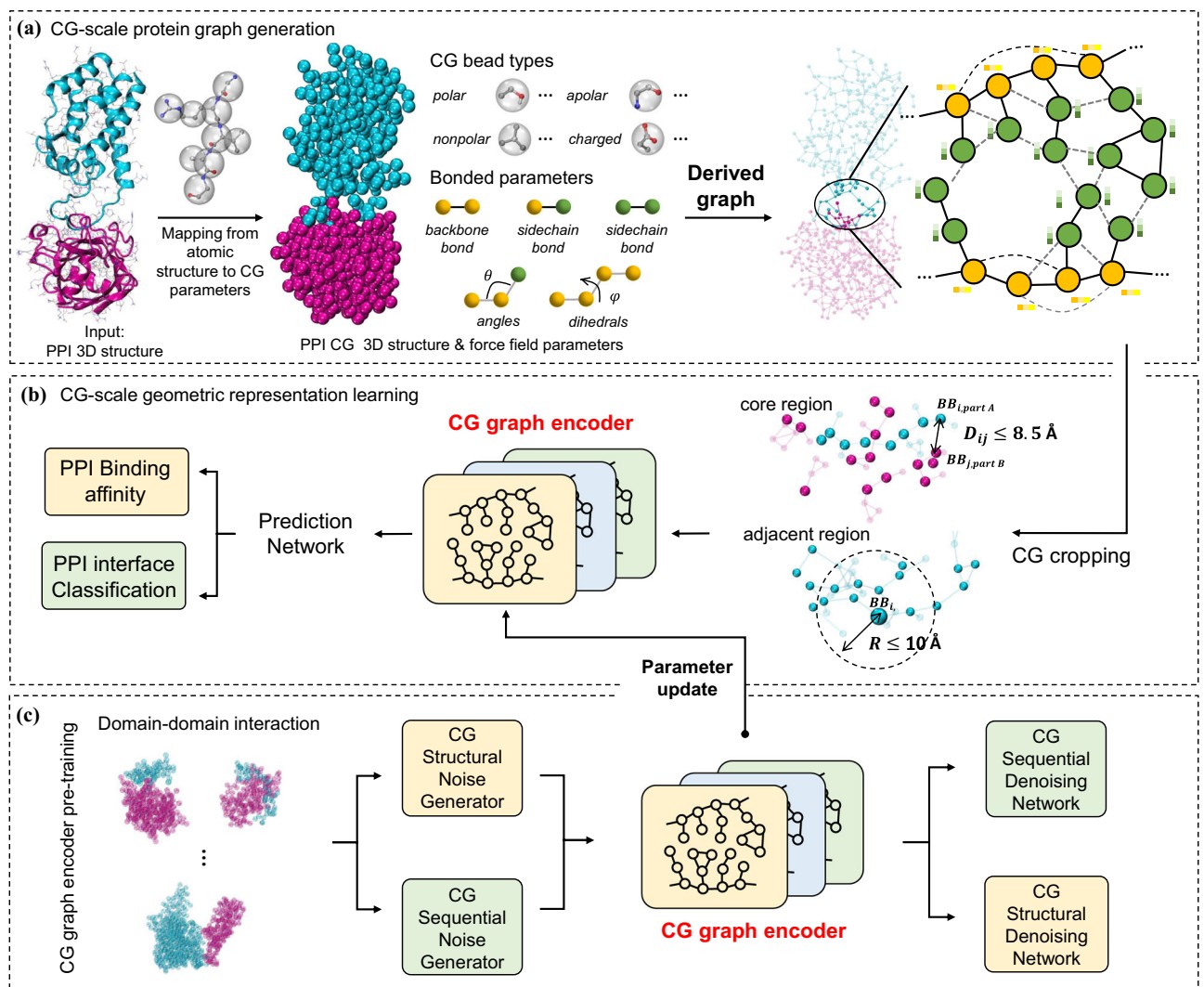

**Fig. 1 | The flowchart of the MCGLPPI framework. a**, CG-scale protein graph generation. The atomistic structure of a protein-protein complex is transformed into its coarse-grained (CG)-scale structure and force field parameters using the MARTINI engine (MARTINI22 and MARTINI3 are supported in this study), with these parameters encompassing bead types and bonded interactions, which further include bonds, angles, and dihedrals. Based on these structural details and parameters, we create a CG-scale protein graph where beads are represented as nodes. Bonds between these beads are represented as edges in the graph, and information on angles and dihedrals is encoded as node features **b**, CG-scale geometric representation learning. The specifically designed CG-scale protein complex

graph, containing comprehensive information of MARTINI beads and bonds, is firstly cropped to identify its core interaction region. The geometric representation of this cropped graph is extracted by the corresponding CG graph encoder, which is then fed into a protein prediction network to predict the overall property of the complex. **c**, DDI-based CG-scale graph encoder pre-training. The graph encoder can be better initialized by the CG-scale pre-training techniques applied to our carefully screened (domain-domain interaction) DDI dataset. After pre-training, the graph encoder with updated model parameters can be fine-tuned to generate geometric representations with potentially more powerful capability for downstream prediction tasks. CG: coarse-grained. PPI: protein-protein interaction.

After integrating the structural data with the force field parameters, a multi-relational graph corresponding to the protein complex is constructed (Figs. 1a and 2). Within this graph, each bead, representing a group of heavy atoms, becomes a node. The bonds between backbone beads ($B$) or between sidechain ($S$) and either sidechain or backbone beads defined by their type and length, are translated into edges that connect these nodes. It is worth noting that these nodes and edges are concise (i.e., their total numbers required to depict a protein complex are relatively lower, the corresponding statistics and further analysis for their effect on saving calculation overhead are provided in Supplementary Table 1), ensuring efficient protein modeling while maintaining chemical accuracy.

Within the MARTINI framework, the protein's secondary structure plays a pivotal role in determining the bead types and associated bond, angle, and dihedral parameters for each residue. For instance,

specific bond types such as constraint bonds $d_{B_iB_{i+1}}(H)$ or long harmonic bonds $d_{B_iB_{i+3}}(E)$ and $d_{B_iB_{i+4}}(E)$ are used for regions designated as helices ($H$) or extended strands ($E$), while other backbone bond parameters $d_{B_iB_{i+1}}(CTS)$ are adopted for irregular secondary structures such as coils, turns, and bends. In our CG-scale complex graph, the edge types also reflect these distinctions, facilitating the accurate description of secondary structural features within the protein complex. Furthermore, two distinct edge types, $d_{intra}$ and $d_{inter}$, are introduced for the differentiation of bead nodes originating from the same or different amino acid residues, providing valuable hierarchical geometric information regarding the spatial arrangement relationships and interactions within and between the residues. Additionally, other crucial force field parameters, such as bead types, bond angles, and dihedrals, are encoded as node features within the graph (as illustrated in Fig. 2). These features are essential for

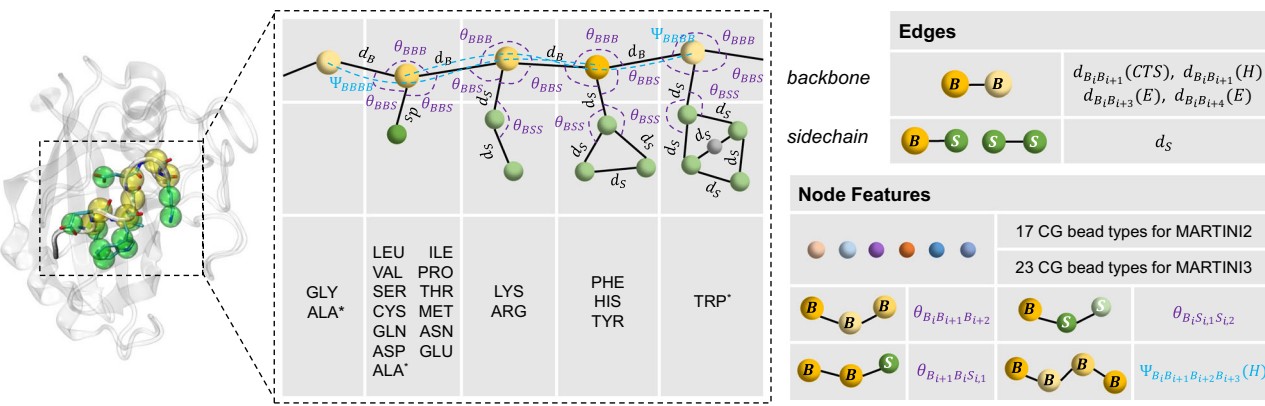

**Fig. 2 | MARTINI-based CG-scale representation of protein structure.** Each residue is represented by one backbone bead ($B$) and zero to five side-chain beads ($S$), depending on the residue type (left) and MARTINI version. The bonding interaction parameters within and between amino acids are shown (to effectively depict their local interaction structures), including bonds (types and lengths), angles (formed by triples of beads), and dihedrals (for quadruplets of beads). The right panel provides an in-depth view of the constructed CG-scale complex graph based on these efficient parameters, showcasing both the edge connections and the node features. It should be noted that in MARTINI2 CG model, ALA does not have side chain beads, whereas in MARTINI3, it has one side chain bead. TRP in MARTINI2 includes four side chain beads, while in MARTINI3, it includes five side chain beads. CG: coarse-grained.

capturing the spatial orientation and potential movements of the protein segments.

Furthermore, when MARTINI generates force field parameters of bond lengths, angles, and dihedrals, what it provides include the bead composition (i.e., from which beads these bonds and angles will form) and values for these bonds lengths and angles. These values are not derived from the corresponding real conformations, instead, they are from the statistical values over samples in the Protein Data Bank (PDB)[25,38] database. To make these values specific to individual parameters for accurate CG-graph construction, we re-calibrate them based on the actual coordinates and give them appropriate feature assignment. Please refer to The construction of CG-scale protein complex graph and its cropping function section in "Methods" section for further details of defining the aforementioned multi-relational graph.

### CG-scale geometric representation learning
To reduce the computational overhead and preserve the integrity of the data for different PPI structures, we implemented a residue backbone distance-based duel-strategy approach to graph cropping on the CG-scale complex graphs derived earlier (Fig. 1b). The initial strategy, core region cropping, focuses on the extraction of interaction interface between two proteins (or defined interaction parts of the structures beyond dimers), to ensure the focus on the most critical region of the interaction, likely enhancing model prediction accuracy and relevance. While the second strategy involves an adjacent region cropping scheme for capturing peripheral but potentially significant structural information like essential spatially correlated motifs surrounding the core interface. Through these strategies, we can produce a graph that balances detailed structural information retention with computational feasibility, regardless of the interaction pattern. The specific cropping details are in The construction of CG-scale protein complex graph and its cropping function section of "Methods" section.

We then apply the cropping method to each complex sample in our curated downstream datasets, which include two types of binding affinity-related regression and one interface type classification tasks. These tasks span a range of complexes, from the formation of simple dimers to the binding of the T cell receptor (TCR) to an antigenic peptide presented by the major histocompatibility complex (pMHC)[39,40].

We subsequently utilize a multi-relational heterogeneous GNN-based CG graph encoder[21], which can efficiently encode the complex relationships between graph nodes and edges (detailed in The CG-scale representation learning for complex overall property prediction section of "Methods" section) within the cropped graph for generating its high-quality geometric representation. This representation is then forwarded to the task-specific prediction network, enabling us to obtain accurate predictions of the corresponding complex overall properties.

### DDI-based CG-scale graph encoder pre-training
Domains are fundamental structural units within proteins that are often responsible for specific functions. They play a critical role in mediating interactions with other proteins[34–36], whether within a single multi-faceted protein (intra-protein interactions) or between two distinct proteins (inter-protein interactions). Despite the limited availability of detailed yet labelled 3D structural data for PPIs, the wealth of DDI structural information provides a valuable opportunity for enhancing computational models through pre-training. To this end, we use the Three-Dimensional Interacting Domains (3DID) database[34] to construct a dataset tailored for pre-training our CG-scale graph encoder. The detailed curation process is described in The detailed curation process for the 3DID pre-training dataset section of "Methods" section.

We employ a denoising-based, self-supervised pre-training approach, adapted from the work by Zhang et al.[22], to instruct our CG graph encoder on the intricate patterns of DDI structures and sequences. This method involves introducing perturbation to each CG graph in the pre-training DDI dataset and then forcing the encoder to reconstruct the original graph information, thereby imprinting the fundamental characteristics of domain interactions (see Fig. 1c and The DDI-based CG graph encoder pre-training technique section in "Methods" section for more details). Following this pre-training phase, the encoder, now enriched with the knowledge from the DDI dataset, undergoes fine-tuning to tackle downstream PPI prediction tasks. Through this fine-tuning process, the encoder applies the principles of domain interactions learned during pre-training to downstream PPI scenarios, potentially enhancing its ability in making predictions.

### MCGLPPI saves computational cost while keeping competitive performance
To validate the performance and computational cost of the proposed MCGLPPI framework on the PPI complex overall property predictions, we first curated three datasets: (1) the strict protein-protein dimer subset of the PDBbind dataset[41] (PDBbind-strict-dimer dataset), (2) the

ATLAS dataset[39], and (3) the MANY/DC dataset[42,43]. The former two datasets were used to evaluate the model's regression capabilities (protein-protein binding affinity predictions), while the MANY/DC dataset was used to assess the overall classification performance (protein complex interface classifications). The Pearson's correlation coefficient ($R_P$), root mean square error (RMSE), and mean absolute error (MAE) were utilized to assess the quality of regressions. The area under the receiver operating characteristic curve (AUROC) and the area under the precision-recall curve (AUPR) were for checking the capability of classification. To ensure a fair comparison, we performed the same aforementioned complex graph cropping function for each sample in every dataset (across different scales) to identify the sample's core interaction regions. Additionally, we managed to 1) compare the performance of MCGLPPI supported by MARTINI22 (denoted as MCGLPPI-M2) and MARTINI3 models (denoted as MCGLPPI-M3) and 2) further extend MCGLPPI into handling protein-protein binding affinity change regressions requiring pairwise complex structures as the input (Supplementary Table 5).

### The binding affinity prediction of the formation of strict dimers

We successfully extracted protein-protein complexes exhibiting strict dimer structures from the PDBbind dataset[41]. Following sample correction and label unification (i.e., converting the binding affinity labels of all relevant samples to $\triangle G$)[44], we obtained 1270 dimer samples with binding affinity labels $\triangle G$, referred to as the PDBbind-strict-dimer dataset (the detailed curation process can be found in Supplementary Note 2). The standard tenfold cross-validation (CV) strategy was used to evaluate the model, specifically, aforementioned sample points will be uniformly split into 10 folds for the CV, in each iteration one fold is selected as the test set, and the rest of folds are treated as the training set. To ensure a fair comparison of the model performance across different scales, we compared the atom- and residue-scale versions of our employed protein graph encoder, GearNet-Edge[21], using their default model settings (i.e., settings related to protein graph construction and geometric encoder hyper-parameters). Besides, we considered an atom-scale state-of-the-art geometric encoder, GVP-GNN[20], specifically designed for solving 3D macromolecular structures, particularly protein-protein complexes. Detailed information on the default hyper-parameters of all methods can be found in Supplementary Note 3.

Furthermore, to comprehensively quantify the cost of these approaches under limited lightweight computational resources, we utilized a single NVIDIA A100 GPU 40GB to run the comparative experiments. For each approach, based on the same epoch number of 150, starting with a batch size of 8 and gradually increased it by a factor of 2 until the GPU was out of memory (OOM), and we recorded the corresponding evaluation metrics, memory usage, and total time cost across the aforementioned tenfold CV.

For the atom- and residue-scale GearNet-Edge, 915 of 1270 samples were successfully identified. To ensure a fair comparison, this 915 subset of the PDBbind-strict-dimer dataset was first used for the comparative experiment. Table 1 presents the corresponding results. Under current experimental conditions, the key findings were as follows: (1) MCGLPPI outperformed its atom- and residue-scale counterparts. (2) Under the same batch size, MCGLPPI reduced GPU consumption by approximately 5× and 3×, as well as total elapsed time by 3× and 3×, compared to the atom- and residue-scale models, respectively, while maintaining competitive performance. These findings demonstrated the effectiveness and feasibility of introducing the MARTINI-based CG-scale representation to achieve a better balance between performance and computational cost. Additionally, MARTINI3-based MCGLPPI had moderately better performance under the best batch size (64) compared to that from MARTINI22, while with slightly increased computation overhead due to the expanding bead types and numbers.

The experimental results of MCGLPPI-M2 and MCGLPPI-M3 on the complete PDBbind-strict-dimer dataset were 0.590/0.583 ($R_P$), 2.071/2.067 (RMSE), 1.602/1.566 (MAE), 11,560/13,311 (GPU (MB)), and 14,312/17,446 (Time (s)). Besides, to validate the robustness of our proposed MCGLPPI, we conducted further investigations into (1) the impact of hyper-parameter, such as hidden feature dimensions on overall performance (Supplementary Note 4), (2) more challenging training and test scenario with further structural homology reduction (Supplementary Table 3), (3) model stability test using AlphaFold-generated structures (Supplementary Table 4). These investigations consistently supported the conclusions outlined above.

### The effectiveness of MCGLPPI on more complex PPI patterns

To further investigate the effectiveness of MCGLPPI in handling more complex PPI structures beyond standard dimers, the ATLAS dataset[39], which contains the TCR-pMHC structures formed in the cell-mediated immunity processes along with their corresponding binding affinity values, was considered. After removing invalid samples, correcting samples, and unifying labels, we obtained 531 different structures with the $\triangle G$ labels. Please note that we utilized the structures that were optimized using the fixed backbone design option of Rosetta[45], which were reported to achieve high structural accuracy[39].

We performed the aforementioned standard tenfold cross-validation using the same experimental settings as the previous section and documented the corresponding evaluation results. Furthermore, the comprehensive comparison experiments were carried out based on 451 of the 531 curated samples that could be effectively processed by GearNet-Edge at both the atom- and residue-scale.

Table 2 shows the predictive performance and computational cost derived from the tenfold cross-validation performed on the 451-sample ATLAS subset. Additionally, we reported the best-performing results of MCGLPPI-M2 and MCGLPPI-M3 on the complete curated ATLAS dataset: 0.809/0.823 ($R_P$), 1.116/1.053 (RMSE), 0.837/0.803 (MAE), 13,615/16,108 (GPU (MB)), and 6982/7915 (Time (s)). Notably, when dealing with more complex protein-protein structures beyond standard dimers, the proposed MCGLPPI maintained the competitive performance and exhibited a relatively lower computational cost compared with its atom- and residue-scale counterparts, which further validated the effectiveness of the devised CG-scale protein complex geometric model and corresponding cropping function. An additional investigation into the necessity of the cropping function is conducted in Influence of graph cropping on overall model efficiency section.

### The prediction results for protein-protein interface classification

In addition to the aforementioned two regression tasks, an overall interface classification task for protein–protein complex was incorporated to further examine the generalizability of MCGLPPI. Specifically, the MANY[42] and DC[43] datasets were utilized, containing 5739 and 161 dimers respectively. These dimers are categorized into two overall types: dimers with biological or crystal interfaces[46]. Based on this classification, the model was trained to distinguish between the two interface types, which was further formulated as a binary complex graph classification task. Following the previous data splitting convention[1,18], 80% samples of MANY, 20% samples of MANY (for MANY dataset splitting, the balance between positive and negative samples were maintained), and the complete DC datasets were used as the training, optional validation, and test sets for model evaluation, respectively.

The experiment settings from the previous two sections were kept (except for the unified epoch number changing from 150 to 30). Additionally, we compared our approach with two existing approaches, DeepRank-GNN[18] and EGGNet[1], which had already been tested on the complete MANY/DC dataset. However, it should be noted that the

**Table 1 | Test performance and computational cost of different approaches at different scales on the 915-subset of the PDBbind-strict-dimer dataset based on one A100 GPU 40GB**

| Batch size = 128 | Scale | $R_P$ | RMSE | MAE | GPU (MB) | Time (s) |
|---|---|---|---|---|---|---|
| MCGLPPI-M2 | CG | 0.584 | 2.103 | 1.623 | **22,098** | **9724** |
| MCGLPPI-M3 | CG | **0.603** | **2.083** | **1.601** | 25,131 | 10,812 |
| GearNet-Atom[22] | Atom | N/A | N/A | N/A | OOM | N/A |
| GearNet-Res[21] | Residue | N/A | N/A | N/A | OOM | N/A |
| GVP-GNN[20] | Atom | N/A | N/A | N/A | OOM | N/A |
| **Batch size = 64** | **Scale** | $R_P$ | **RMSE** | **MAE** | **GPU (MB)** | **Time (s)** |
| MCGLPPI-M2 | CG | 0.597 | 2.065 | 1.595 | **11,366** | **11,228** |
| MCGLPPI-M3 | CG | **0.603** | **2.063** | **1.581** | 13,337 | 12,987 |
| GearNet-Atom[22] | Atom | N/A | N/A | N/A | OOM | N/A |
| GearNet-Res[21] | Residue | N/A | N/A | N/A | OOM | N/A |
| GVP-GNN[20] | Atom | N/A | N/A | N/A | OOM | N/A |
| **Batch size = 32** | **Scale** | $R_P$ | **RMSE** | **MAE** | **GPU (MB)** | **Time (s)** |
| MCGLPPI-M2 | CG | 0.587 | **2.100** | **1.619** | **5969** | **12,247** |
| MCGLPPI-M3 | CG | **0.590** | 2.109 | 1.621 | 6850 | 13,159 |
| GearNet-Atom[22] | Atom | 0.527 | 2.320 | 1.788 | 30,597 | 35,993 |
| GearNet-Res[21] | Residue | 0.578 | 2.182 | 1.664 | 21,295 | 35,336 |
| GVP-GNN[20] | Atom | 0.436 | 2.299 | 1.795 | 24,639 | 44,174 |
| **Batch size = 16** | **Scale** | $R_P$ | **RMSE** | **MAE** | **GPU (MB)** | **Time (s)** |
| MCGLPPI-M2 | CG | **0.585** | **2.085** | **1.615** | **3381** | **12,817** |
| MCGLPPI-M3 | CG | 0.583 | 2.108 | 1.632 | 3888 | 13,484 |
| GearNet-Atom[22] | Atom | 0.541 | 2.283 | 1.755 | 17,229 | 47,169 |
| GearNet-Res[21] | Residue | 0.578 | 2.149 | 1.642 | 10,544 | 38,703 |
| GVP-GNN[20] | Atom | 0.428 | 2.369 | 1.839 | 13,424 | 46,155 |
| **Batch size = 8** | **Scale** | $R_P$ | **RMSE** | **MAE** | **GPU (MB)** | **Time (s)** |
| MCGLPPI-M2 | CG | 0.571 | **2.109** | 1.636 | **1853** | **15,796** |
| MCGLPPI-M3 | CG | 0.559 | 2.144 | 1.652 | 2230 | 17,355 |
| GearNet-Atom[22] | Atom | 0.541 | 2.372 | 1.821 | 8476 | 50,545 |
| GearNet-Res[21] | Residue | **0.574** | 2.110 | **1.624** | 7140 | 43,382 |
| GVP-GNN[20] | Atom | 0.424 | 2.383 | 1.853 | 7199 | 48,355 |

The default model settings of atom-scale GearNet-Edge (here denoted as GearNet-Atom), residue-scale GearNet-Edge (denoted as GearNet-Res), and (atom-scale) GVP-GNN were adopted from refs. 22,21,20, respectively.
The bold data signifies the best experimental result under the current batch size and evaluation metric. The data with the underline represents the best predictive performance according to the current evaluation metric across different batch sizes. *OOM* out of memory for one A100 40GB/40,000MB GPU. *N/A* not applicable.

effective sample numbers for atom- and residue-scale GearNet-Edge on the MANY and DC datasets were 5535 and 151, respectively. Moreover, the node feature construction in existing approaches like DeepRank-GNN relies on time-consuming external amino acid sequence alignment search, making it difficult to fairly compare computational cost. Therefore, we only compared their predictive performance on the complete MANY/DC dataset and conducted detailed computational cost comparison experiments for the atom- and residue-scale GearNet-Edge models on the 5535-151-sample subset (following the aforementioned data splitting mode).

The results of the computational cost comparison experiments are shown in Table 3. It was observed that compared to its atom- and residue-scale counterparts, MCGLPPI achieved lower computational cost while surpassing their predictive capability. Specifically, MCGLPPI-M2 and MCGLPPI-M3 exhibited strong performance when evaluated with a batch size of 64, achieving AUROC values of 0.890 and 0.882, respectively. Additionally, the AUPR values for these models were 0.871 and 0.881, respectively. Overall, both models outperformed the performance of atom-scale and residue-scale models across different batch sizes. The reason for this improvement could be attributed to the integration of protein thermodynamics and specific secondary structure support information through the MARTINI force field, which is injected into the bonds (edges) of the CG complex

graph, which provides extra distinguishable capability compared with its atom- and residue-scale counterparts. A further investigation into the importance of different CG graph edges are presented in Performance of the geometries considered in CG-scale complex graphs section.

Furthermore, the performance of MCGLPPI supported by MARTINI22 (AUROC: 0.895, AUPR: 0.892) also outperformed both DeepRank-GNN (AUROC: 0.865, AUPR: 0.871) and EGGNet (AUROC: 0.869, AUPR: 0.863) on the complete MANY/DC dataset (the results of DeepRank-GNN and EGGNet were retrieved from the refs. 1,18). This observation further supported the effectiveness and generalizability of the MCGLPPI framework for predicting overall properties of PPI complexes.

## The investigation of CG-scale pre-training techniques on different tasks

To explore the feasibility of our hypothesis that pre-training on CG-scale informative DDI complexes could benefit the downstream property predictions of CG complexes, especially in scenarios with limited labeled samples, we constructed a dataset from the 3DID database[34], which comprises 41,663 representative DDI structures, serving as our pre-training repository. Furthermore, we implemented a CG-scale diffusion denoising-based self-supervised pre-training

**Table 2 | Test performance and computational cost of different approaches at different scales on the 451-subset of the curated ATLAS dataset based on one A100 GPU 40GB**

| Batch size = 128 | Scale | $R_P$ | RMSE | MAE | GPU (MB) | Time (s) |
|---|---|---|---|---|---|---|
| MCGLPPI-M2 | CG | **0.807** | **1.084** | **0.833** | **26,358** | **5578** |
| MCGLPPI-M3 | CG | 0.803 | 1.121 | 0.861 | 31,514 | 6378 |
| GearNet-Atom[22] | Atom | N/A | N/A | N/A | OOM | N/A |
| GearNet-Res[21] | Residue | N/A | N/A | N/A | OOM | N/A |
| GVP-GNN[20] | Atom | N/A | N/A | N/A | OOM | N/A |
| **Batch size = 64** | **Scale** | $R_P$ | **RMSE** | **MAE** | **GPU (MB)** | **Time (s)** |
| MCGLPPI-M2 | CG | 0.825 | 1.054 | **0.782** | **13,285** | **5915** |
| MCGLPPI-M3 | CG | <u>**0.832**</u> | **1.013** | 0.783 | 15,919 | 6500 |
| GearNet-Atom[22] | Atom | N/A | N/A | N/A | OOM | N/A |
| GearNet-Res[21] | Residue | N/A | N/A | N/A | OOM | N/A |
| GVP-GNN[20] | Atom | N/A | N/A | N/A | OOM | N/A |
| **Batch size = 32** | **Scale** | $R_P$ | **RMSE** | **MAE** | **GPU (MB)** | **Time (s)** |
| MCGLPPI-M2 | CG | 0.820 | 1.061 | 0.778 | **6796** | **5992** |
| MCGLPPI-M3 | CG | **0.825** | <u>**1.009**</u> | <u>**0.756**</u> | 8066 | 7281 |
| GearNet-Atom[22] | Atom | 0.813 | 1.109 | 0.851 | 31,018 | 21,375 |
| GearNet-Res[21] | Residue | 0.818 | 1.068 | 0.796 | 21,059 | 19,866 |
| GVP-GNN[20] | Atom | 0.524 | 1.599 | 1.169 | 26,232 | 27,231 |
| **Batch size = 16** | **Scale** | $R_P$ | **RMSE** | **MAE** | **GPU (MB)** | **Time (s)** |
| MCGLPPI-M2 | CG | 0.821 | 1.042 | 0.788 | **3524** | **7225** |
| MCGLPPI-M3 | CG | **0.831** | **1.017** | **0.757** | 4251 | 8034 |
| GearNet-Atom[22] | Atom | 0.816 | 1.062 | 0.772 | 16,155 | 22,579 |
| GearNet-Res[21] | Residue | 0.820 | 1.056 | 0.816 | 11,314 | 22,276 |
| GVP-GNN[20] | Atom | 0.534 | 1.494 | 1.111 | 13,211 | 30,027 |
| **Batch size = 8** | **Scale** | $R_P$ | **RMSE** | **MAE** | **GPU (MB)** | **Time (s)** |
| MCGLPPI-M2 | CG | 0.818 | 1.048 | 0.801 | **1892** | **8790** |
| MCGLPPI-M3 | CG | 0.812 | **1.046** | **0.773** | 2345 | 9447 |
| GearNet-Atom[22] | Atom | **0.823** | 1.106 | 0.813 | 8255 | 26,810 |
| GearNet-Res[21] | Residue | 0.809 | 1.086 | 0.832 | 6190 | 24,157 |
| GVP-GNN[20] | Atom | 0.480 | 1.573 | 1.170 | 6946 | 32,253 |

The default model settings of atom-scale GearNet-Edge (here denoted as GearNet-Atom), residue-scale GearNet-Edge (denoted as GearNet-Res), and (atom-scale) GVP-GNN were adopted from refs. 22,21,20, respectively.
The bold data signifies the best experimental result under the current batch size and evaluation metric. The data with the underline represents the best predictive performance according to the current evaluation metric across different batch sizes. *OOM* out of memory for one A100 40GB/40,000MB GPU. *N/A* not applicable.

technique based on ref. 22, which enables us to capture and learn the general DDI patterns and knowledge.

Specifically, we first determined the optimal model settings for MCGLPPI under each downstream task through training-from-scratch separately. Using the same selected settings (for each task), we fine-tuned the CG graph encoder that had undergone pre-training for each respective downstream task (with the same epoch numbers as training-from-scratch). We then compared the performance difference between training-from-scratch and pre-training with fine-tuning. Moreover, to further explore the potential of the DDI dataset-based pre-training in enhancing PPI prediction models, we extended our performance comparison to the atom-scale and residue-scale (i.e., pre-training them using the original-scale corresponding pre-training settings[22] based on the 3DID pre-training set). It should be noted that the atom- and residue-scale models (i.e., GearNet-Edge) could only recognize 33,144 out of the 41,663 DDI samples. Therefore, we selected these 33,144 samples as the common pre-training set and transformed these samples into their respective atom-, residue-, and CG-scale graphs to compare the performance across different scales (before and after pre-training) (Fig. 3a).

In the PPI binding affinity prediction tasks on the PDBbind and ATLAS datasets, taking MCGLPPI-M2 as an example, pre-training improved the $R_P$ from 0.597 to 0.606 and from 0.825 to 0.830,

respectively, indicating that pre-training can effectively enhance model performance. However, for the interface type classification task on the MANY/DC dataset, the performance actually decreased after pre-training, with AUPR dropping from 0.880 to 0.866 (additional results evaluated using other metrics and those tested on the complete 3DID dataset for MCGLPPI were reported in Supplementary Table 2). Meanwhile, the consistent performance change trend was found on MCGLPPI-M3, GearNet-Atom, and GearNet-Res.

We thought the contributing reasons are as follows. For the binary classification task that aims to distinguish between biological interfaces and crystal artefacts (representing non-biological interactions), training-from-scratch with current graph settings might be sufficient for the protein learning model to capture the subtle geometric structural difference between these interface types. Furthermore, pre-training based on DDI complexes extracted from actual (biological) protein-protein interactions would be more beneficial to tasks which predict the properties of complexes formed through real PPI processes (e.g., binding affinity predictions), rather than distinguishing crystallographic interfaces resulting from non-biological interactions detected between repetitive crystal units[18].

In addition, we can conclude that, when using the same number of pre-training samples, our CG-scale approaches exhibited the overall better performance on all the PDBbind, ATLAS, and MANY/DC

**Table 3 | Test performance and computational cost of different approaches at different scales on the 5535-151-subset of the MANY/DC dataset based on one A100 GPU 40GB**

| Batch size = 128 | Scale | AUROC | AUPR | GPU (MB) | Time (s) |
|---|---|---|---|---|---|
| MCGLPPI-M2 | CG | **0.875** | **0.880** | **25,437** | **584** |
| MCGLPPI-M3 | CG | 0.873 | 0.855 | 29,172 | 656 |
| GearNet-Atom[22] | Atom | N/A | N/A | OOM | N/A |
| GearNet-Res[21] | Residue | N/A | N/A | OOM | N/A |
| GVP-GNN[20] | Atom | N/A | N/A | OOM | N/A |
| **Batch size = 64** | **Scale** | **AUROC** | **AUPR** | **GPU (MB)** | **Time (s)** |
| MCGLPPI-M2 | CG | <u>**0.890**</u> | 0.871 | **13,269** | **617** |
| MCGLPPI-M3 | CG | 0.882 | <u>**0.881**</u> | 15,150 | 704 |
| GearNet-Atom[22] | Atom | N/A | N/A | OOM | N/A |
| GearNet-Res[21] | Residue | N/A | N/A | OOM | N/A |
| GVP-GNN[20] | Atom | N/A | N/A | OOM | N/A |
| **Batch size = 32** | **Scale** | **AUROC** | **AUPR** | **GPU (MB)** | **Time (s)** |
| MCGLPPI-M2 | CG | **0.883** | **0.876** | **6737** | **702** |
| MCGLPPI-M3 | CG | 0.866 | 0.838 | 8248 | 777 |
| GearNet-Atom[22] | Atom | 0.842 | 0.824 | 34,452 | 3055 |
| GearNet-Res[21] | Residue | 0.848 | 0.842 | 24,368 | 3031 |
| GVP-GNN[20] | Atom | 0.850 | 0.845 | 27,467 | 4132 |
| **Batch size = 16** | **Scale** | **AUROC** | **AUPR** | **GPU (MB)** | **Time (s)** |
| MCGLPPI-M2 | CG | 0.860 | 0.849 | **3952** | **951** |
| MCGLPPI-M3 | CG | **0.882** | **0.864** | 4417 | 1091 |
| GearNet-Atom[22] | Atom | 0.850 | 0.799 | 19,290 | 3293 |
| GearNet-Res[21] | Residue | 0.844 | 0.843 | 13,505 | 3208 |
| GVP-GNN[20] | Atom | 0.817 | 0.808 | 15,506 | 4515 |
| **Batch size = 8** | **Scale** | **AUROC** | **AUPR** | **GPU (MB)** | **Time (s)** |
| MCGLPPI-M2 | CG | **0.881** | 0.857 | **2229** | **961** |
| MCGLPPI-M3 | CG | 0.877 | **0.867** | 2548 | 1065 |
| GearNet-Atom[22] | Atom | 0.839 | 0.758 | 10,026 | 4017 |
| GearNet-Res[21] | Residue | 0.870 | 0.859 | 8578 | 3661 |
| GVP-GNN[20] | Atom | 0.836 | 0.825 | 8427 | 5194 |

The default model settings of atom-scale GearNet-Edge (here denoted as GearNet-Atom), residue-scale GearNet-Edge (denoted as GearNet-Res), and (atom-scale) GVP-GNN were adopted from refs. 22,21,20, respectively.
The bold data signifies the best experimental result under the current batch size and evaluation metric. The data with the underline represents the best predictive performance according to the current evaluation metric across different batch sizes. *OOM* out of memory for one A100 40GB/40,000MB GPU. *N/A* not applicable.

datasets. In total, pre-trainings on DDIs are effective in enhancing PPI predictions, with the CG-scale graph encoder combined with CG DDIs pre-training being the most effective approach.

## Performance of the geometries considered in CG-scale complex graphs

To assess the impact of various geometric representations within CG complex graphs, based on the relatively simple and classical MARTINI22-based MCGLPPI, we conducted a series of ablation studies. These experiments were designed to gradually remove specific graph components and analyze their individual contributions to the overall predictive performance of the system.

Initially, we focused on edges based on chemically plausible interactions as defined by the MARTINI(22) force field. These interactions included various types of bonds between beads, such as $d_{B_i B_{i+1}}(CTS)$, $d_{B_i B_{i+1}}(H)$, $d_{B_i B_{i+3}}(E)$, $d_{B_i B_{i+4}}(E)$, and $d_S$ (detailed in The construction of CG-scale protein complex graph and its cropping function section of "Methods" section, and the following analyzed graph components can be referred from the same section). We used the subset of 915 protein dimers from the PDBbind-strict-dimer dataset for our analysis. Upon selective removal of all MARTINI bond-based edges from the CG graphs, we observed a measurable decline in performance metrics. The $R_P$ decreased from 0.597 to 0.569 (Fig. 3b),

confirming the importance of these edges in accurately characterizing protein interactions.

Next, after removing MARTINI bond-based edges, we investigated the effectiveness of the proposed bead-residue geometric hierarchical composition-aware edges $d_{intra}$ and $d_{inter}$. We replaced these two types of edges with the standard radius-based edges that do not differentiate the compositional relationships between chemically-plausible bead nodes and their corresponding residues (using the same edge cutoff). This modification resulted in a further decrease in predictive accuracy, with $R_P$ dropping from 0.569 to 0.557 (Fig. 3b). This emphasized the significant impact that composition-aware edges have on the model.

We also evaluated scenarios within a complete CG graph where, if both residue composition-aware edges and MARTINI bond-based edges are present between the same pair of end nodes, the MARTINI bond-based edges are disregarded. Under these conditions, the predictive performance experienced a decline from an $R_P$ of 0.597 to 0.573 (Fig. 3b). This result emphasized the importance of explicitly incorporating MARTINI bond-based edges along with the composition-aware edges in our modeling framework.

Furthermore, other groups of MARTINI force field parameters, such as bead types, angles ($\theta_{B_i B_{i+1} B_{i+2}}$, $\theta_{B_{i+1} B_i S_{i,1}}$, $\theta_{B_i S_{i,1} S_{i,2}}$), and dihedrals

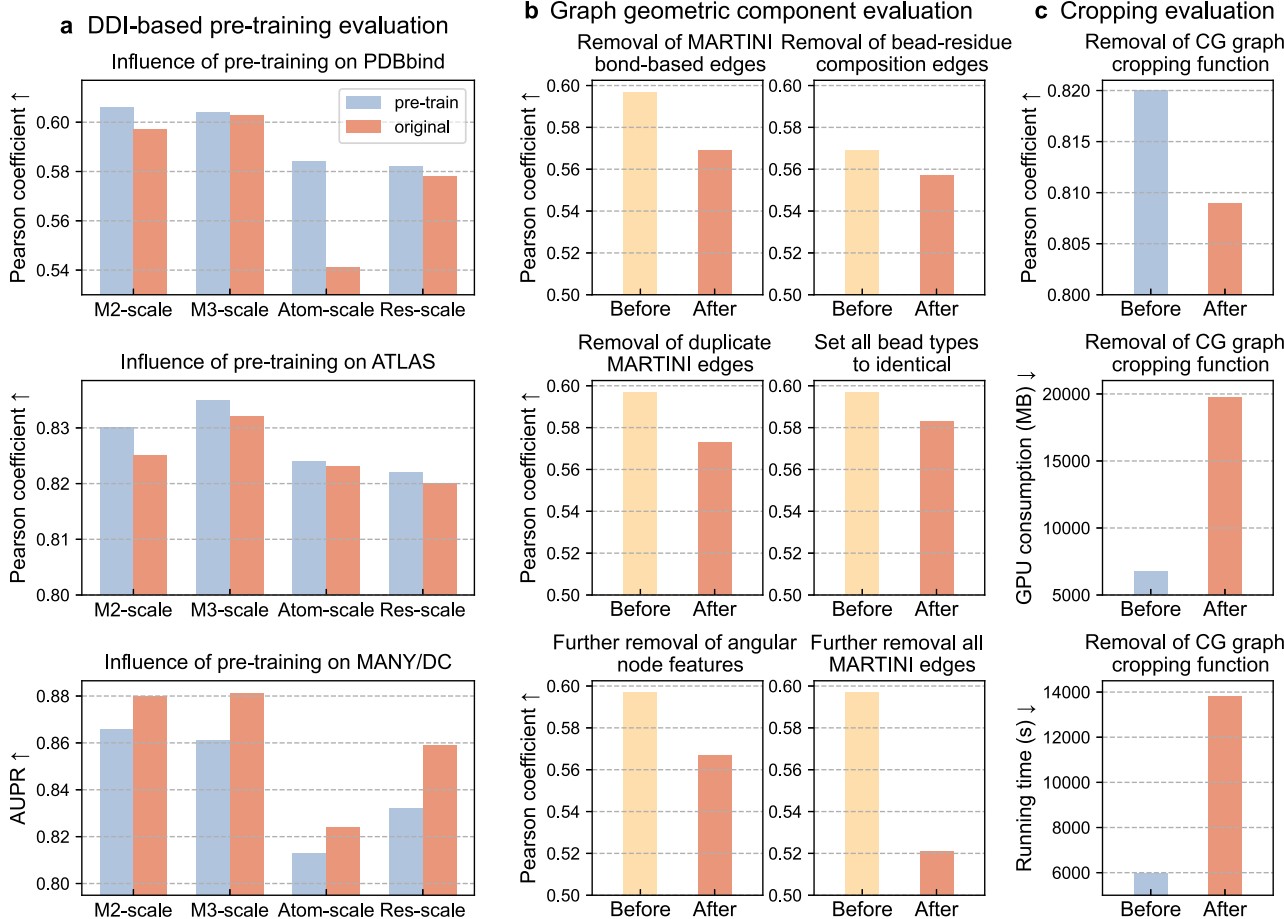

**Fig. 3 | The investigation of performance influence contributed by the DDI-based pre-training, crucial graph geometric characterization components, and cropping function of MCGLPPI. a**, Performance influence brought by the DDI diffusion denoising-based pre-training on the three downstream datasets. "M2-scale", "M3-scale", "Atom-scale", "Res-scale" represent the MARTINI22 CG-scale pre-training on the 3DID 33144-set, MARTINI3 CG-scale pre-training on the 33144-set, atom-scale pre-training on the 33144-set, and residue-scale pre-training on the 33144-set, respectively; and the blue and red bars display the results obtained after and before imposing the pre-training, respectively. **b**, Performance influence of removing different geometric characterization components from the devised MCGLPPI-M2 complex graph. The respective experiments about the removal of

different components (based on PDBbind) are indicated on top of each sub-figure, and the yellow and red bars give the results acquired before and after removing the corresponding components, respectively. **c**, Performance influence related to the complex cropping function. Based on the maximum reachable batch size of MCGLPPI(-M2) without the complex cropping function (i.e., 32), the performance comparison results of MCGLPPI before (blue bars) and after (red bars) closing the cropping function based on the ATLAS dataset, including the $R_P$, GPU consumption, and total tenfold CV running time are provided. CG: coarse-grained. ↑ and ↓ represent the higher/lower the metric value, the better the evaluation performance. Source data are provided as a Source Data file.

($\Psi_{B_i B_{i+1} B_{i+2} B_{i+3}}$), were encoded as node or edge features within the graph. To assess their specific importance, we invalidated the bead type feature and angular features, which include (bond) angles and dihedrals, from the CG-scale graph. As shown in Fig. 3b, when the bead type in every node feature and edge feature were set to identical or when angular information was further omitted (on top of the former), there was a reduction in the $R_P$ from 0.597 to 0.583 and 0.567, respectively.

Moreover, the complete invalidation of above two groups of force field parameters plus all MARTINI-based edges (i.e., only the standard radius-based edges are included), led to a further decline in $R_P$ to 0.521 (reaching a 12.7% difference). These findings indicated that, not only does our framework rely on aforementioned chemically-plausible edges, but it also requires both the chemical and physical information provided by the MARTINI bead types and angular information from angles and dihedrals. This combined information enables MCGLPPI to make more informed predictions about the properties of protein interactions.

## Influence of graph cropping on overall model efficiency

To provide a comprehensive analysis of the influence of graph cropping on the efficiency of the MCGLPPI framework, we conducted additional experiments using the ATLAS dataset, which contains more complex TCR-pMHC structures as a benchmark (based on MARTINI22-version MCGLPPI). We ran our model on the subset of 451 protein complex structures from the ATLAS dataset while maintaining all experimental settings consistent with our previous MCGLPPI experiments, except for the graph cropping function, which was disabled. As shown in Fig. 3c, the maximum batch size that could be processed by MCGLPPI under a single NVIDIA A100 GPU 40GB decreased significantly from 128 to 32 when the graph cropping function was turned off. With a batch size of 32, the $R_P$, GPU memory consumption, and total runtime post-cropping were 0.825, 6796 MB, and 5992 s, respectively, compared to 0.809, 19,706 MB, and 13,794 s before cropping. These results confirmed the critical role of graph cropping in improving computational efficiency and predictive performance in the framework.

## Discussion

In this study, we presented MCGLPPI, an efficient framework that enhances the structure-based overall property predictions for protein-protein complexes by utilizing the MARTINI force field for lightweight protein modeling. At the mesoscopic CG-scale, our proposed CG protein graph model uses concise yet chemically plausible beads and bonds to accurately represent the conformation characteristics of protein-protein complexes. This approach results in lower computational overhead, leading to a better balance between predictive performance and cost compared to its atom- and residue-scale counterparts. Further, the modern design of protein graphs and corresponding GNN protein encoders like GearNet-Edge is more relying on the construction of edges, which are usually fully built based on multiple pre-defined geometric distance and sequential thresholds, aiming to capture more comprehensive spatial relationships between particle nodes[21,22,47]. While the number of edges will significantly influence the neighboring message aggregation[17] speed of corresponding GNNs. Our devised CG-graph incorporates more chemical-plausible MARTINI-based edges wiring designed bead nodes pairs based on specific interaction definitions, thereby reducing the reliance on indiscriminately connecting every node pair within multiple pre-defined thresholds, which is ultimately beneficial to decreasing the processing overhead under current framework (more detailed explanation can be found in Supplementary Table 1).

On top of this, MCGLPPI was designed to adapt to the geometric parameters from both the classical MARTINI22 and the more recent MARTINI3 force field for flexible extensions and comparisons. Under current experimental settings, different versions achieve better performance on different datasets, while the DDIs-based pre-training narrows down the difference between the two versions. Besides, MARTINI22-based MCGLPPI enjoys slightly quicker processing speed due to less bead types and bead numbers (please see more specific performance difference analysis across versions in Supplementary Note 5).

In addition, our extensive ablation studies highlighted the significance of both edge and node features derived from the MARTINI force field for accurate PPI predictions using the MCGLPPI framework. Notably, these features, like chemical bond-based edges and physical-plausible bead type node features, are crucial in capturing the essential properties that govern protein interactions (The potential extension analysis of using the force field parameters in other scales can be found in Supplementary Note 6). Through our proposed CG-scale learning framework, we also demonstrated the effectiveness of DDI-based pre-training in improving binding affinity predictions of PPIs.

While MCGLPPI has shown promising overall performance, there are some areas for further improvement or investigation. For instance, MCGLPPI may not fully capture the complexity of protein-protein systems, and other CG-scale protein modeling systems deserve further exploration. Besides, the MCGLPPI model, built on a geometric GNN framework, learns from the confident 3D structures of complexes to predict related overall properties. Although with simple modifications, we further demonstrated its versatility by straightforwardly extending it to binding affinity change (i.e., $\triangle\triangle G$) calculation of pairwise wildtype-mutant complexes on an independent multiple-point amino acid mutation dataset (detailed in Supplementary Table 5), it currently lacks the capability to (directly) utilize the more broadly available and readily accessible PPI sequence data as initial input of predicting PPI attributes.

Based on this, we plan to incorporate more cost-efficient geometric information to more comprehensively characterize CG complex structures, e.g., considering the Euler angles to describe the relative rotation between CG particles, and support more CG modeling systems to capture protein-protein thermodynamic quantities and underlying chemical mechanisms from different perspectives. Furthermore, there is a potential to further improve model performance on more PPI tasks by integrating sequence co-evolutionary information as a feature component[48]. Additionally, combining our CG-scale framework (with further adaptive modifications) more intensively with tools that can predict confident PPI structures based on sequences, such as AlphaFold3[49], AlphaFold-Multimer[50], FoldDock[51], or the MARTINI force field-integrated HADDOCK[31], will open avenues for predicting PPI properties, such as determining whether two proteins interact and better understanding the effects of mutations on these interactions.

## Methods

### The detailed curation process for the 3DID pre-training dataset

The latest 3DID database[34] provides 15,983 DDI structure templates, with each template containing one or more samples of resolved 3D structural data. We removed any DDI templates from the 3DID dataset that were identical to those present in our downstream benchmark datasets. Following this stringent exclusion process, we obtained a pre-training dataset which provides 41,663 DDI structure samples in total.

### MARTINI-based geometric parameter generation

Each complex structure was processed using pdbfixer tool (https://github.com/openmm/pdbfixer) to complete missing side-chain information and convert non-natural amino acids to their natural counterparts (for the atom- and residue-scale models, the same process was also performed). The Python-based MARTINI script martinize.py (https://cgmartini.nl/docs/downloads/tools/proteins-and-bilayers.html, version 2.4)[23] was used to generate MARTINI22-based CG structure and force field parameters for each protein complex. The Martinize2 and Vermouth programs[52] were for producing MARTINI3-based CG structure and force field parameters of corresponding complexes. The CG structure encompasses the bead coordinate-related information, while the force field parameters include both nonbonded parameters, such as bead types, and bonded parameters, including bonds, angles, dihedrals, and bead connectivity instructing the bead composition for these bonds and angles (Fig. 2).

**MARTINI-based bonds information.** Within the MARTINI (applicable for both MARTINI22 and MARTINI3) force field representation, there are two principal types of bonds: backbone bonds and sidechain bonds. As illustrated in Figs. 1a and 2, a backbone bond $d_B$ is formed between two neighboring backbone beads ($B_i$). Sidechain bonds $d_S$ occur either between a backbone bead and a sidechain bead ($S_i$) within the same amino acid or between sidechain beads. Furthermore, the MARTINI force field differentiates backbone bond types based on the protein's secondary structure. Consequently, $d_B$ can be subdivided into:

1. Constraint bonds: These are formed between two adjacent amino acids that are part of a helical structure ($H$), denoted as $d_{B_iB_{i+1}}(H)$.
2. Long harmonic backbone bonds: For three consecutive amino acids forming extended elements ($E$), these bonds connect the backbone beads of residues $i$ and $i+3$, denoted as $d_{B_iB_{i+3}}(E)$.
3. Long harmonic backbone bonds: For four contiguous amino acids in extended elements ($E$), the bonds connect backbone beads of residues $i$ and $i+4$, denoted as $d_{B_iB_{i+4}}(E)$.
4. Other harmonic backbone bonds: Parameters between two adjacent amino acids for irregular secondary structures such as coils, turns, and bends are denoted as $d_{B_iB_{i+1}}(CTS)$.

In total, there are five types of bonds in both MARTINI22 and MARTINI3 force fields: $d_{B_iB_{i+1}}(CTS)$, $d_{B_iB_{i+1}}(H)$, $d_{B_iB_{i+3}}(E)$, $d_{B_iB_{i+4}}(E)$, and $d_S$. These bond types were chosen as the primary edge types in our MCGLPPI framework.

**MARTINI-based angles and dihedrals information.** The bonded parameters also include angle and dihedral parameters (Fig. 2).

Specifically, there are three types of angle parameters and one dihedral type being considered:

1. $\theta_{B_iB_{i+1}B_{i+2}}$: The angle between three consecutive backbone beads.
2. $\theta_{B_{i+1}B_iS_{i,1}}$: The angle formed between a backbone bead, its neighboring backbone bead, and the first sidechain bead of the amino acid.
3. $\theta_{B_iS_{i,1}S_{i,2}}$: The angle between a backbone bead and two consecutive sidechain beads of the same amino acid.
4. $\Psi_{B_iB_{i+1}B_{i+2}B_{i+3}}$: The dihedral angle between four consecutive backbone beads. It is noted that dihedral angles were imposed only when all four interacting beads had the helical secondary structure (H) in MARTINI force field.

Notably, the bond lengths, angle and dihedral values were recalibrated based on the given coordinates of corresponding endpoint beads. They were not directly adopted from the statistics values from PDB database since the calibration can provide accurate geometric interactive information for the CG-scale protein complex graph model.

## The construction of CG-scale protein complex graph and its cropping function

The challenge to build an effective CG protein complex graph is how to fully preserve the introduced MARTINI parameters in a graph structure accurately and efficiently, while keeping the flexibility of injecting other useful knowledge on top of these MARTINI parameters. To overcome this challenge, we first modelled a given protein-protein complex as a multi-relational contact graph $\mathscr{G} = (\mathcal{V}, \mathcal{E}, \mathcal{R})$. $\mathcal{V}$ represents the set of graph nodes $i$, i.e., all MARTINI beads produced for the complex, and the position of each bead node $i$ is determined by its equipped 3D coordinate. $\mathcal{E}$ and $\mathcal{R}$ are the set of edges between bead nodes and the set of edge types $r$, respectively. Based on this, we denoted an edge from nodes $j$ to $i$ with type $r$ as $(i, j, r)$).

In order to give the precise descriptions of inter-bead geometric positional relationships while integrating concise chemical-plausible MARTINI bonds, the graph structure is built as follows. First, an edge will be wired if any two bead nodes have the Euclidean distance smaller than 5Å. Compared with the commonly-used radius edge used in atom- and residue-scale models, MCGLPPI further distinguishes the edge type based on whether the two end bead nodes are from the same residue. In other words, the edge is categorized as an intra-residue contact edge $d_{intra}$ if the two bead nodes belong to the same residue otherwise is an inter-residue contact edge $d_{inter}$, for which the hierarchical composition information between chemical-plausible beads and their surrounding residues within a complex is injected. Besides, all bond types described in the MARTINI-based bonds information section are incorporated in the graph edge structure.

To summarize, there are seven types of edges in total, describing the protein complex geometry from different perspectives, including various precise geometric contact relationships, actual secondary structure supports, and chemical bonded interactions, etc. Although multiple types of edges are included, due to the conciseness of MARTINI bonds, the average degree of nodes in the constructed graph is still relatively small (see Supplementary Table 1 for further comparison analysis with the atom- and residue-scale counterparts), which contributes to relatively low representation learning overhead. Next, the other MARTINI parameters were allocated as follows into these defined nodes and edges as their features (the definition of generation of a one-hot representation is in Supplementary Note 7).

### Bead node features $\mathbf{f}_i$:

- The MARTINI22 or MARTINI3 bead type, given as a one-hot representation
- Sine-cosine encoded backbone angles ($[\sin(\theta_{BBB}), \cos(\theta_{BBB})]$)
- Sine-cosine encoded backbone-side chain angles ($[\sin(\theta_{BBS}), \cos(\theta_{BBS})]$)
- Sine-cosine encoded side chain angles ($[\sin(\theta_{BSS}), \cos(\theta_{BSS})]$)
- Sine-cosine encoded backbone dihedrals ($[\sin(\Psi_{BBBB}), \cos(\Psi_{BBBB})]$)

### Graph edge features $\mathbf{f}_{(i,j,r)}$:

- The one-hot MARTINI22 or MARTINI3 bead type of the source node
- The one-hot MARTINI22 or MARTINI3 bead type of the target node
- The edge type, given as a one-hot representation
- The absolute positional difference between source and target nodes in the MARTINI22 or MARTINI3 bead sequence, given as a one-hot representation
- The calibrated bond length

For the above individual node and edge features, they were concatenated as the final features ($\mathbf{f}_i : \mathbf{R} \in 1 \times 25$ (MARTINI22) or $1 \times 31$ (MARTINI3), $\mathbf{f}_{(i,j,r)} : \mathbf{R} \in 1 \times 53$ (MARTINI22) or $1 \times 65$ (MARTINI3)). Besides, since the angular parameters generated from MARTINI are sparse (see Fig. 2, not every bead will involve in the calculation of every type of angles), an extra rule (Supplementary Note 8) was provided, to assign sparse angular node features to specific beads to avoid potential conflicts.

After the definition of the CG-scale complex graph, the corresponding graph cropping function was designed to identify its core interaction regions for further reducing the computational cost and potentially increasing the predictive accuracy. Specifically, we first determined the interaction parts of the protein-protein complexes with different interaction patterns. For the curated complexes in the PDBbind and MANY/DC datasets, they belong to the standard dimers, we chose each protein chain as one part, thus a complex can be represented as two interaction parts. For the structures in ATLAS which usually contain 4 or 5 chains, the peptide and MHC chains were treated as the first part, while the second part contained the remained TCR chain structures.

Next, a distance matrix $\mathbf{M}^{dis}$ based on the specified two interaction parts was created to guide the generation of the cropped complex graph $\mathscr{G}'$. For each residue, MARTINI will only assign one backbone bead ($B$) with a 3D coordinate to represent its backbone atoms and their overall position, and thus the coordinate of $B$ was used as the position of the residue (analogous to using alpha carbon ($C\alpha$) as the residue position in residue-level protein graph constructions[21]). Based on this, $\mathbf{M}^{dis}$ with the size of $L_{AA1} \times L_{AA2}$ can be calculated, in which $L_{AA1}$ and $L_{AA2}$ are the amino acid (AA) sequence length (given by above backbone beads $B$) of the interaction parts 1 and 2, respectively. In $\mathbf{M}^{dis}$, every element $\mathbf{M}^{dis}_{i,j}$ was the pairwise Euclidean distance between corresponding residues from separate interaction parts (given by $B_i$ and $B_j$). Then any pair of residues having the distance smaller than 8.5Å were retained as the core region (i.e., the initial strategy). Other AAs that had $B - B$ distance to any core region AAs smaller than 10Å were also retained (i.e., the second strategy).

After that, all bead nodes and edges within the retained region were kept as the cropped complex graph $\mathscr{G}'$ (the angular node features were further re-calibrated if any end bead nodes within the angular information calculations were removed by this cropping). Furthermore, for the atom-scale and residue-scale models, to conduct a fair comparison, the same cropping function were employed to their protein graphs, the only difference was replacing $B$ with $C\alpha$ to indicate the overall position of the residue.

## The CG-scale representation learning for complex overall property prediction

After acquiring the cropped protein complex graph $\mathscr{G}'$, a representative multi-relational heterogeneous GNN-based protein encoder GearNet-Edge[21], was incorporated into the framework,

for predicting the overall property of protein complexes. Specifically, based on a line graph-enhanced edge message passing mechanism[53] to model the inter-edge positional relationships, the additional structural information can be injected into the node representations for more effective protein geometric interaction modeling (the corresponding equations are provided in Supplementary Note 9). We made it work at the CG-scale for generating the overall geometric representation of the input CG cropped graph $\mathcal{G}'$, and the generated representation was further learnt by a three-layer task-specific multi-layer perception (MLP) to give the final property prediction results.

**The DDI-based CG graph encoder pre-training technique**
The protein domain-domain complex parameterized by MARTINI still preserves the fundamental conformation and chain sequence, but the basic particles are substituted from original atoms to CG beads. Intuitively, performing the self-supervised noise-adding-denoising pre-training techniques, which are already demonstrated to be effective on understanding geometric regulations of proteins at atom- or residue-scale[54], could also benefit the understanding of general knowledge from CG DDI complexes (for downstream property predictions).

Therefore, based on the atom-scale work[22], a CG-scale complex pre-training technique was developed, which adds noise with changing magnitudes into 3D coordinates and sequences of MARTINI-based CG bead nodes based on the diffusion mechanisms[55], for the CG-complex geometric regulation learning. The equations and complete details are provided in Supplementary Note 10. After the pre-training, the trained CG graph encoder will be fine-tuned on the specified downstream task to produce the effective structural representations for corresponding input CG complex graphs.

For the implementation of the CG-scale protein complex graph construction, representation learning, and pre-training processes, Pytorch[56] and Torchdrug[57] with a default random seed 0 were employed, and the Adam[58] with the initial learning rate of 0.0001 was adopted as the optimizer for model training (the same environment settings were also used for the other involved models working at the atom- and residue-scale). Besides, all experiments were deployed on a configuration of one NVIDIA A100 GPU 40GB. A complete summary of tools for implementing MCGLPPI is given in Supplementary Note 11.

**Reporting summary**
Further information on research design is available in the Nature Portfolio Reporting Summary linked to this article.

## Data availability
All datasets analyzed are freely available through the original sources: (1) 3DID: https://3did.irbbarcelona.org/, (2) PDBbind: http://www.pdbbind.org.cn/, (3) ATLAS: https://pubmed.ncbi.nlm.nih.gov/28160322/, (4) MANY and DC: https://www.eppic-web.org/ewui/#downloads, and (5) AB-bind: https://www.ncbi.nlm.nih.gov/pmc/articles/PMC4815335/. Easing access, we re-packaged all data at https://github.com/arantir123/MCGLPPI. When using those data, please quote and consult the authors of the original datasets. Source data are provided within this paper. Source data are provided with this paper.

## Code availability
The source code of MCGLPPI (Version 1.0) can be downloaded from https://github.com/arantir123/MCGLPPI.

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

## Acknowledgements

This work was supported by National Key Research and Development Program of China (2022YFF1202104), National Natural Science Foundation of China (62471310), and National Natural Science Foundation of China (32341002, 32030035). This work was also supported by the Computer Science Ramsay Fund at the University of Birmingham (to Y.Y.). We are thankful for the financial support from Macao Polytechnic University Foundation (RP/FCA 07/2022 to S.L.). We also thank Dr. D. McDonald, Dr. M. Heinzinger, Ms. C. Marquet, and Dr. B. Rost for their helpful suggestions on our task studies. The computations described in this research were performed using the Baskerville Tier 2 HPC service. Baskerville was funded by the EPSRC and UKRI (EP/T022221/1 and EP/W032244/1) and is operated by Advanced Research Computing at the University of Birmingham.

## Author contributions

Y.Y. and S.L. conceived the research project. L.W., T.H., Z.Z., and S.H. supervised the research project. Y.Y. designed the computational pipeline. Y.Y. implemented the MCGLPPI framework and performed the model training and prediction validation tasks. S.L. curated all involved protein complex samples and conducted experiments for MARTINI force field-based geometric parameter generation. Y.Y., S.L., Y.C., L.W., T.H., Z.Z, and S.H. wrote the manuscript.

## Competing interests

The authors declare no competing interests.
