## [Transparent Peer Review file · Nature Communications]

Integration of molecular coarse-grained model into geometric representation learning framework for protein-protein complex property prediction

Corresponding Author: Dr Shan He

Version 0:

Reviewer comments:

Reviewer #1

(Remarks to the Author)

An interesting paper, describing how PPI can be classified more efficiently using a ML framework based on the coarse-grain Martini force field.

Overall the presented results look convincing, and a number of additional tests are provided to reveal the relevance of the various input features as well as importance of pre-training the network based on domain-domain interactions.

I do have some serious concerns that warrant further discussion or explanation:

1) It is unclear why the proposed CG-based method outperforms a residue-based approach. The rationale presented in the beginning of the manuscript is that a CG representation of the protein will be more accurate than a residue-based one, but apparently it is also faster. This requires further explanation, as a residue-based graph in principle is simpler and therefore should be faster.

2) The choice for the Martini 22 model requires further argumentation. First of all, why did the authors not use the more recent Martini 3 version? Second, examples (from the literature) should be provided that demonstrate the successful use of Martini in predicting protein-protein complexes. In the current manuscript it is only mentioned that Martini is successful in reproducing transfer free energies of amino acids. While this might be true, it is of less relevance of predictive capabilities of PPI. More relevant would be studies focussing on amino acid dimerization (e.g. de Jong et al. JCTC 2012) and protein dimerization (e.g. Lamprakis et al. JCTC 2021). Also of interest is the integration of the Martini model into the Haddock workflow to predict PPI (Roel-Touris et al. JCTC 2019).

3) Representing Martini proteins with a graph is not new, and underlies for instance the martinize program that was also used in this study to construct the CG models. It is strange that this is not being mentioned - the program itself is not even cited (Kroon et al., eLife 2023).

4) The authors mention on lines 389-391 that the improvement of their model is caused by integration of thermodynamic data and specific structural data of Martini as features into their GNN. The question is if you would be able to get the same improvement when such information was used in the all-atom or residue based networks? Or is this only accessible with Martini? If so, please explain why.

5) It appears that many features could be left out without compromising the accuracy of the predictions too much. R_p values only slightly go down, even when for instance all the bead type information is left out. This leaves me wondering what the added benefit of using a force field such as Martini really is - it appears like the protein-protein interfacial properties could be largely determined just considering the number of contacts, and not taking into account the chemical specific information. Some discussion on this aspect would strengthen the paper.

Some minor issues:

- 6) First subheader of Results section ('Overall of MCGLPPI') is too cryptic.
- 7) A number of times the authors mention that nodes/edges are 'concise', it is not clear what is being meant by this wording.
- 8) Lines 197-202 are unclear. Why is additional calibration required? How is that linked to statistical analysis of the PDB?
- 9) What is 'one-hot' representation?

Reviewer #2

(Remarks to the Author)

The article "Integration of molecular coarse-grained model into geometric representation learning framework for protein-protein complex property prediction" by Yang Yue, Shu Li, Yihua Cheng, Zexuan Zhu, Lie Wang, Tingjun Hou and Shan He presents a new framework for the prediction of protein-protein interactions using a coarse grained (MARTINI) representation.

The approach presented by the authors is legitimate and very interesting. The results on benchmarks confirm that the presented method attains the objective of increasing speed compared to all-atom or residue level representations without loss of accuracy which is a clear validation of the strategy.

This reviewer is however concerned with the choice of the Martini2 version rather than the more recent Martini3. In particular Martini 2 that is well known for being too 'sticky' which pushed the development of Martini 3. Martini 3 also has two specific advantages that should be beneficial in this case: i) it allows to vary the grain of the beads, ii) it includes more chemical types which could be important for the design of PPI inhibitors.

Taking all that into account, Martini3 might have been a better choice at least intuitively. It could be, however, that the pretraining performed by the authors compensate for the difference between both versions of the Martini force field. The ideal would be to test the framework with Martini3 although it represents a huge amount of work. If not possible, the rationale for the choice and potential implications should at least be thoroughly discussed.

Reviewer #3

(Remarks to the Author)

This paper introduces a novel graph network based on the Martini models to predict binding strength (and other properties) from a known protein-complex structure. The method seems to perform on par (or slightly better) than residue and atom-based methods and has a substantial computational advantage.

I have some more fundamental problems and some more technical problems with the article. These are listed (in random order) below.

1. First, one can wonder how significant the problem is. What you want to study is not the ΔG of protein-protein interaction (in particular for a structure that is known) but rather the $\Delta\Delta G$ for a mutation in the interface. The authors should (at the bare minimum) try the method on the ProteinGym (the subset for interactions) and compare it with state-of-the-art methods for $\Delta\Delta G$ predictions.
2. How well does this (and the other method) on models of PPIs (easily obtained by AlphaFold). Are they stable?
3. How important is computational efficiency? Getting a protein structure costs \$1000s so if you really need to predict the ΔG you can probably afford to spend some compute on it. For models (and large-scale all-against-all)
4. How well does the method compare with methods based on co-evolution (Topsy Turvy, etc) or combined folding-docking (FoldDock AF-multimer) for separating interacting-non interacting methods
5. The paper does not explain how a method using ~ 3 atoms/residue can be more computational efficient than a residue-based method (using 1 atom/residue). I would guess that this has to do with some cutoffs for the graphs or something similar. But then the question comes: Are these optimized equally for both representations?
6. No homology reduction was used (from what I found) in the cross-validation. This results in a severe risk for over-training.
7. The paper is rather long and talkative but lacks details on testing/training set etc.

/arne

Version 1:

Reviewer comments:

Reviewer #1

(Remarks to the Author)

The authors addressed my concerns well, and I am happy to recommend publication of this interesting work.

Reviewer #2

(Remarks to the Author)

The authors have successfully taken into account suggestions made by me and other reviewers making the article more robust and, in my opinion, ready for publication.

Reviewer #3

(Remarks to the Author)

As pointed out in my initial review, the ability to predict ΔG is not of great interest, and I seriously doubt that the improvements obtained here will be of any significance when you consider the experimental accuracies. Even trivial methods, such as measuring the buried hydrophobic surface area are likely to do quite well on predicting ΔG .

Therefore, I would again insist that the author demonstrate that their method can be used to predict $\Delta\Delta G$ values before publishing this paper. If that is not possible that hints at this method not being useful for the scientific community.

Version 2:

Reviewer comments:

Reviewer #3

(Remarks to the Author)

I appreciate the authors' attempts to predict $\Delta\Delta G$ - but in contrast to what they try to claim in this paper, this hints at the low value of the initial predictions. If the initial method actually was useful, there would be no need to use a "three-layer MLP" to predict the $\Delta\Delta G$ from the hWT-hMT. It would just be possible to use $\Delta\Delta G = \Delta G_{WT} - \Delta G_{MT}$ directly without any training. I assume that the authors tried that, but the results were too bad to mention - which is scientifically misleading. Further, it is not clear how homology reduction or cross-validation was performed in this test - it is not clear from Ref 26 if they did use homology reductions on the antigens or not (they should have but from what I can see it is not described in that way).

Manuscript ID: NCOMMS-24-15723

Manuscript Title: Integration of molecular coarse-grained model into geometric representation learning framework for protein-protein complex property prediction

Response to Reviewer 1

We are very grateful for your comments and thoughtful suggestions. We have carefully considered all comments and have revised the manuscript accordingly. All revisions are highlighted in blue for easy identification.

Q1 It is unclear why the proposed CG-based method outperforms a residue-based approach. The rationale presented in the beginning of the manuscript is that a CG representation of the protein will be more accurate than a residue-based one, but apparently it is also faster. This requires further explanation, as a residue-based graph in principle is simpler and therefore should be faster.

A1 Authors' Response:

The modern design of residue-based graphs and corresponding graph neural network (GNN) protein encoders like GearNet-Edge [1-3] for protein property predictions is more relying on the construction of edges, which are usually fully built based on multiple pre-defined geometric distance or sequential thresholds, aiming to capture more comprehensive spatial relationships between residue nodes [1-3]. While the number of edges will significantly influence the neighboring message aggregation [4] speed of corresponding graph neural networks. For the proposed coarse-grained (CG)-based method MCGLPPI, we incorporated more chemical-plausible MARTINI force field-based edges. For these MARTINI-based edges, they are constructed based on the specific interaction definitions between designated bead node pairs, thereby reducing the reliance on indiscriminately wiring the every node pair within multiple pre-defined thresholds (and important interaction information is still kept).

We calculated the node and edge number related information for the CG-scale graphs and their residue-scale counterparts [1, 2] in the MANY/DC dataset [5, 6] used in our study as an example (Supplementary Table 1). Although under current settings, the average graph node number in CG-scale is relatively higher, the average graph edge number and node degree are both observed an significant decrease, which is ultimately beneficial to decreasing the its calculation overhead and increasing the processing speed under current framework.

We have discussed the aforementioned content about the efficiency of the CG-scale graph and residue-scale graph in Discussion section of the manuscript (see pages 27-28, lines 557-568). We have also provided more detailed information for graph size across different scales in “The statistics of graph nodes and edges of the MANY/DC dataset” section of Supplementary Information on pages 2-4, lines 37-68.

Q2 The choice for the Martini 22 model requires further argumentation. First of all, why did the authors not use the more recent Martini 3 version? Second, examples (from the literature) should be provided that demonstrate the successful use of Martini in predicting protein-protein complexes. In the current manuscript it is only mentioned that Martini is successful in reproducing transfer free energies of amino acids. While this might be true, it is of less relevance of predictive capabilities of PPI. More relevant would be studies focusing on amino acid dimerization (e.g. de Jong et al. JCTC 2012) and protein dimerization (e.g. Lamprakis et al. JCTC 2021). Also of interest is the integration of the Martini model into the Haddock workflow to predict PPI (Roel-Touris et al. JCTC 2019).

A2 Authors' Response:

We appreciate the reviewer's question about the choice of the MARTINI force field. Initially, our study employed the MARTINI22 model due to its established reliability and simplicity with its parameters. In 2021, the MARTINI developers released the MARTINI3 force field, which offers improved interaction balance and enhanced accuracy in biomolecular simulations. In the latest revised version of our manuscript, we have also integrated the MARTINI3 force field with our computational framework, referred to as MCGLPPI-M3. Detailed comparison results and further discussions regarding this integration can be found across the revised manuscript.

Additionally, we thank the reviewer for the suggestion on providing some successful works in PPI predictions by using the MARTINI model and providing some referenced works. We have introduced these important works on page 5, lines 96-102 of the manuscript:

“It has been successfully applied and evaluated in many PPI-related studies, including the dimerization of the amino acid side chains [7], interactions involving membrane proteins such as glycoprotein A [8], G protein-coupled receptor (GPCR) rhodopsin [9, 10], and the Epidermal Growth Factor Receptor complex [11], as well as soluble protein complex interactions [10] like insulin, Ras-Raf, and Barnase-Barstar. Additionally, the MARTINI force field has been integrated into the HADDOCK framework [12], enhancing its capability to predict the 3D structures of protein interactions.”

Q3 Representing Martini proteins with a graph is not new, and underlies for instance the martinize program that was also used in this study to construct the CG models. It is strange that this is not being mentioned - the program itself is not even cited (Kroon et al., eLife 2023).

A3 Authors' Response:

In this study, we used a Python-based script called martinize.py (downloaded from <http://www.cgmartini.nl/index.php/tools2/proteins-and-bilayers/204-martinize>) to convert the atomistic structure to the MARTINI22-based CG structure and its corresponding force field parameters. The

Martinize2 program is showcased as a generalization of the martinize script, allowing us to use a broader range of CG force fields and simulation engines [13]. We used the Martinize program to generate the MARTINI3 CG structures and topologies. Detailed information can be found on page 30, lines 622-627 of the manuscript.

Q4 The authors mention on lines 389-391 that the improvement of their model is caused by integration of thermodynamic data and specific structural data of Martini as features into their GNN. The question is if you would be able to get the same improvement when such information was used in the all-atom or residue based networks ? Or is this only accessible with Martini ? If so, please explain why.

A4 Authors' Response:

We thank the reviewer for the question. In principle, similar thermodynamic and structural data could be integrated into all-atom or residue-based models. However, integrating all-atom force field-based parameters into the corresponding GNN model is computationally expensive, and using residue-based force field parameters makes it challenging to capture detailed interactions accurately. The MARTINI force field provides a unique balance between computational efficiency and accuracy by reducing the number of interaction sites, which is particularly advantageous for our GNN-based framework.

We have carefully discussed the integration of thermodynamic data and structural information from different force fields into the corresponding scale GNN models in “The potential extension of using corresponding force field parameters in other scales” of Supporting Information on pages 9-10, lines 183-206.

Q5 It appears that many features could be left out without compromising the accuracy of the predictions too much. R_p values only slightly go down, even when for instance all the bead type information is left out. This leaves me wondering what the added benefit of using a force field such as Martini really is - it appears like the protein-protein interfacial properties could be largely determined just considering the number of contacts, and not taking into account the chemical specific information. Some discussion on this aspect would strengthen the paper.

A5 Authors' Response:

Thank you for your suggestion. In the experiments related to the subgraphs of Fig. 3b “Set all bead types to identical” and “Identical bead type with no angular features”, the bead type information actually exists in two parts of features, i.e., CG graph node features and edge features (please see the “The construction of CG-scale protein complex graph and its cropping function” subsection of Methods for detailed description). The original purpose of these experiments was to examine the performance influence of removing bead type information from node features (and corresponding information in edge features is

still remained).

To make the overall ablation study more clear and ultimately better demonstrate the benefit of adding these different groups of force field parameters, we conducted three updated comparisons on top of the previous ones. The first one is keeping other experimental settings unchanged, invalidating the bead type information from both node and edge features. The second one is removing angular features in node features based on the setting of the first one. The third one is, on top of the second one, further removing the force field component – all MARTINI-based edges, and only the commonly-used radius edges are included (see Methods). The change of R_p brought by these settings was recorded.

We observed that compared with the complete feature setting in MCGLPPI, R_p decreases from 0.597 to 0.583, 0.567, and 0.521 (reaching a 12.7% difference), respectively, which shows more significant performance influence compared with the aforementioned original comparison experiments that only removed bead type information from the node features. This observation also indicates that these groups of MARTINI force field feature components added to our framework can effectively contribute to increasing the predictive accuracy.

We have updated the aforementioned comparison results into the subgraphs of Fig. 3b and into the corresponding description of the main text (pages 25-26, lines 518-533).

Q6 First subheader of Results section ('Overall of MCGLPPI') is too cryptic.

A6 Authors' Response:

We have changed the first sub-header of Results section to “Overall of the proposed CG-scale complex geometric learning framework” to make it more clear (page 7, line 132).

Q7 A number of times the authors mention that nodes/edges are 'concise', it is not clear what is being meant by this wording.

A7 Authors' Response:

The meaning of word “concise” used to describe our CG-scale nodes and edges is that, they can use relatively lower node and edge numbers to effectively describe the chemical properties of the protein complexes. We have given this definition to the first place where it occurs in the main text of manuscript (pages 9-10, lines 192-195).

Q8 Lines 197-202 are unclear. Why is additional calibration required? How is that linked to statistical analysis of the PDB?

A8 Authors' Response:

We thank the reviewer for pointing out this clarity issue. We have added the following description into the pages 10-11, lines 213-220 of the manuscript to explain this:

“Furthermore, when MARTINI generates force field parameters of bond lengths, angles, and dihedrals, what it provides include the bead composition (i.e., from which beads these bonds and angles will form) and values for these bonds lengths and angles. These values are not derived from the corresponding real conformations, instead, they are from the statistical values over samples in the Protein Data Bank (PDB) [14, 15] database. To make these values specific to individual parameters for accurate CG-graph construction, we re-calibrate them based on the actual coordinates and give them appropriate feature assignment. Please refer to the Methods section for further details.”

Q9 What is 'one-hot' representation?

A9 Authors' Response:

“One-hot” representation refers to transforming the categorical scalar value from current feature (e.g., bead type of MARTINI22 or MARTINI3) into a unique binary representation, with the size equaling to the number of total categories appearing in current feature (e.g., 17 for bead type of MARTINI22 or 23 for bead type of MARTINI3). In this representation, the position corresponding to the current category is marked as 1, while all other positions are marked as 0.

We have added this description into the pages 10-11, lines 207-213 of Supplementary Information.

Response to Reviewer 2

We are very grateful for your comments and thoughtful suggestions. We have carefully considered all comments and have revised the manuscript accordingly. All changes made to the text are in blue so that they may be easily identified.

Q1 This reviewer is however concerned with the choice of the Martini2 version rather than the more recent Martini3. In particular martini 2 that is well known for being too ‘sticky’ which pushed the development of Martini 3. Martini 3 also has two specific advantages that should be beneficial in this case: i) it allows to vary the grain of the beads, ii) it includes more chemical types which could be important for the design of PPI inhibitors.

Taking all that into account, Martini3 might have been a better choice at least intuitively. It could be, however, that the pretraining performed by the authors compensate for the difference between both versions of the Martini force field.

The ideal would be to test the framework with Martini3 although it represents a huge amount of work. If not possible, the rationale for the choice and potential implications should at least be thoroughly discussed.

A1 Authors’ Response:

We thank the reviewer for this thoughtful suggestion. Following the suggestion from the reviewer, we additionally implemented a MCGLPPI framework which also supports the force field parameters from the MARTINI3 version (denoted as MCGLPPI-M3 in the manuscript), and added MCGLPPI-M3 into every comparison of the main experiments in the manuscript.

Under current experimental settings, MCGLPPI powered by different MARTINI versions achieve better performance on different datasets. For example, when training from scratch, in the task of predicting protein-protein interaction (PPI) affinities, MCGLPPI-M3 achieves moderately better performance under the tenfold cross-validation on the PDBbind and ATLAS datasets. At the same time, the DDIs-based pre-training narrows down the difference between the two versions. In addition, MARTINI22-based MCGLPPI enjoys slightly quicker processing speed due to less bead types and bead numbers.

We have updated all the aforementioned contents into the whole manuscript and provided more detailed performance difference analysis across different MARTINI versions in Supplementary Information (please see pages 8-9, lines 159-182 of Supplementary Information).

Response to Reviewer 3

We are very grateful for your comments and thoughtful suggestions. We have carefully considered all comments and have revised the manuscript accordingly. All changes made to the text are in blue so that they may be easily identified.

Q1 First, one can wonder how significant the problem is. What you want to study is not the delta-G of protein-protein interaction (in particular for a structure that is known) but rather the ddG for a mutation in the interface. The authors should (at the bare minimum) try the method on the ProteinGym (the subset for interactions) and compare it with state-of-the-art methods for ddG predictions.

A1 Authors' Response:

We appreciate the reviewer's suggestion and understand the importance of evaluating our method in the context of dG predictions for protein-protein interactions (PPIs). In this study, we proposed an innovative CG-scale MCGLPPI framework, which aims to enhance the efficiency and applicability for various PPI property prediction tasks compared to the atom-scale and residue-scale counterparts. We have demonstrated the predictive capability of the MCGLPPI framework for PPI dG on the PDBbind and ATLAS datasets, and showcased its application in PPI structure classifications on the MANY/DC dataset. The dG value reveals the binding strength between two proteins, helping quantify the intensity and specificity of these interactions. dG can be utilized to effectively explore the binding strength between different proteins. Meanwhile, ddG reflects the impact of amino acid mutations on PPIs. It requires the model to learn the pairwise structural differences before and after the mutation to predict the ddG value [16]. However, our current MCGLPPI framework was specifically designed to support the single (complex) structure input to predict related overall properties and cannot handle energy changes due to mutations in PPIs. This is a limitation of our model for PPI tasks and a direction for future research.

Additionally, ProteinGym [17] is a comprehensive dataset that mainly provides information on the impact of single protein mutations, primarily based on sequence information. Although ProteinGym offers a rich set of mutation samples, its data mainly focus on the effects of mutations in individual proteins. Our study is more focused on the PPI (structure), designing a novel CG-scale-based framework to efficiently predict PPI-related overall properties based on PPI structural information.

We have discussed the aforementioned limitation of our framework in Discussion section of the manuscript (please see pages 28-29, lines 590-596, 603-609 of the manuscript).

Q2 How well does this (and the other method) on models of PPIs (easily obtained by AlphaFold). Are they stable?

A2 Authors' Response:

Following the reviewer's suggestion, we have conducted an additional experiment about the model stability test using more accessible AlphaFold-generated structures.

We first generated the protein complex structures for samples in the PDBbind dataset using the ColabFold (a simplified version based on AlphaFold [18] and AlphaFold-Multimer [19]) [20], then transformed the

generated structures into the CG-graphs and their atom-scale and residue-scale counterparts, and the commonly-identified sample set was selected. Based on this sample set, we performed the performance comparison based on two different sample splitting settings. The experimental results illustrate that, under current experimental settings, the models trained with more accessible AlphaFold-generated structures achieve close predictive accuracy compared with those derived from real structures, indicating that when feeding the AlphaFold-generated structures, the models are also capable of adapting to their structural patterns to give reasonable predictions, which is valuable when the expensive real structures are difficult to acquire.

The detailed information can be found in “Model stability test using more accessible AlphaFold-generated structures” section of Supporting Information on pages 12-14, lines 237-266.

Q3 How important is computational efficiency? Getting a protein structure costs \$1000s so if you really need to predict the dG you can probably afford to spend some compute on it. For models (and large-scale all-against-all).

A3 Authors' Response:

We thank the reviewer for highlighting the importance of computational efficiency. Although obtaining a complex structure experimentally is indeed costly, there are now more and more structure prediction methods available that can computationally generate confident protein structures quickly and at a lower cost based on sequence information. Tools such as AlphaFold2 [18], RoseTTAFold [21] can predict single protein structures from sequences efficiently. Moreover, the latest methods like AlphaFold3 [22], AlphaFold-Multimer [19], and FoldDock [23] can predict PPI complex structures from protein sequences. Additionally, traditional docking tools such as HADDOCK [12], ZDOCK [24], and HDOCK [25], as well as the latest ML-based docking tools like EquiDock [26], can quickly obtain complex structures by inputting either monomer structures or sequences.

Our research also showed that the MCGLPPI framework performed well in terms of both efficiency and accuracy when predicting PPI dG values for complex structures generated using the ColabFold (please see the corresponding description in the last question). This highlights the framework's potential for integrating with these advanced structure prediction tools to provide reliable and efficient PPI property predictions for large-scale studies (either based on real structures or computational generated structures).

Q4 How well does the method compare with methods based on co-evolution (Topsy Turvy, etc) or combined folding-docking (FoldDock AF-multimer) for separating interacting-non interacting methods.

A4 Authors' Response:

Thank you for your suggestion. The discussion regarding sequence co-evolution information and other

PPI structure prediction methods, such as FoldDock and AF-Multimer, for predicting PPI-related properties has been carefully addressed on page 29, lines 590-596 of the manuscript:

“The MCGLPPI model, built on a geometric GNN framework, learns from the single confident 3D structure of complex to predict related overall properties. However, it currently lacks the capability to directly utilize the more broadly available and readily accessible PPI sequence data as initial input for predicting PPI attributes.”

and on page 29, lines 601-609 of the manuscript, we further elaborate:

“Furthermore, there is a potential to further improve model performance on more PPI tasks by integrating sequence co-evolutionary information as a feature component [27]. Additionally, combining our CG-scale framework (with further adaptive modifications) more intensively with tools that can predict confident PPI structures based on sequences, such as AlphaFold3 [22], AlphaFold-Multimer [19], FoldDock [23], or the MARTINI force field-integrated HADDOCK [12], will open new avenues for predicting PPI properties, such as determining whether two proteins interact and understanding the effects of mutations on these interactions.”

Q5 The paper does not explain how a method using ~3 atoms/residue can be more computational efficient than a residue-based method (using 1 atom/residue). I would guess that this has to do with some cutoffs for the graphs or something similar. But then the question comes: Are these optimized equally for both representations?

A5 Authors' Response:

Thank you for your suggestion. Achieving higher computational efficiency in CG-scale is not related to the cutoff for the graphs and is mainly contributed by our design of edges in the CG graph.

At first, for the CG graph and residue graph, they share the same cutoff when cropping the graph to identify the core region of the interaction. The graph cropping function is based on the pairwise Euclidean distance between residues from different interaction chains or parts, which is thus irrelevant to the usage of CG- or residue-scale (for details please check “The construction of CG-scale protein complex graph and its cropping function” subsection in Methods).

In terms of the design of graph edges, the modern design of residue-based graphs and corresponding graph neural network (GNN) protein encoders like GearNet-Edge [1-3] for protein property predictions is more relying on the construction of edges, which are usually fully built based on multiple pre-defined geometric distance or sequential thresholds, aiming to capture more comprehensive spatial relationships between residue nodes [1-3]. While the number of edges will significantly influence the neighboring message aggregation [4] efficiency of corresponding graph neural networks. For the proposed CG-scale graph, we incorporated more chemical-plausible MARTINI force field-based edges. For these

MARTINI-based edges, they are constructed based on the specific interaction definitions between designated bead node pairs, thereby reducing the reliance on indiscriminately wiring the every node pair within multiple pre-defined thresholds (and important interaction information is still kept).

We calculated the node and edge number related information for the CG-scale graphs and their residue-scale counterparts [1, 2] in the MANY/DC dataset [5, 6] used in our study as an example (Supplementary Table 1). Although under current settings, the average graph node number in CG-scale is relatively higher, the average graph edge number and node degree are both observed an significant decrease, which is ultimately beneficial to decreasing the its calculation overhead and increasing the processing speed under current framework.

We have discussed the aforementioned content about the efficiency of the CG-scale graph and residue-scale graph in Discussion section of the manuscript (see pages 27-28, lines 557-568).

Q6 No homology reduction was used (from what I found) in the cross-validation. This results in a severe risk for over-training.

A6 Authors' Response:

Based on the PDBbind dataset, we conducted an additional more strict experiment following the suggestion of the reviewer. Specifically, for the PDBbind 915-subset for which all involved methods can completely identified, we performed the pairwise TM-align structural alignments [28] against all samples in this 915-subset, resulting in a 915×915 TM-score matrix with each element representing the structural similarity between the corresponding samples. Furthermore, for each sample, we chose the maximum TM-score in the corresponding row of the matrix, i.e., its maximum structural similarity against all other samples in the dataset. On top of this, we extracted samples with maximum TM-score lower than 0.45 as the test set, representing the samples with the lowest homology structure similarities (with other all samples) in the dataset (124 samples in total). The samples with the TM-score ranging from 0.45-0.55 were used as the validation set (85 samples in total), and the rest of samples were treated as the training set (706 samples).

Based on the aforementioned homology reduction experimental settings, we selected the best basic model settings under the corresponding tenfold cross-validation (CV) settings for each involved method (including those with and without pre-training), to conduct the comparison (see Supplementary Table 3). The experimental results indicate that, MCGLPPI brings significant performance improvements on Pearson's correlation coefficient (R_p) under this challenging test scenario, further demonstrating the generalization ability of the proposed framework (please see pages 11-12, lines 214-236 of Supplementary Information).

Q7 The paper is rather long and talkative but lacks details on testing/training setc etc.

A7 Authors' Response:

We have tried to modify and remove the unnecessary description from the main text of manuscript. We have also provided more details related to the training and test sets in the manuscript (please see page 14, lines 289-293, page 18, lines 356-358, and page 19, lines 381-384 of the manuscript).

References

1. Zhang Z, Xu M, Jamasb A R, et al. Protein Representation Learning by Geometric Structure Pretraining[C]//The Eleventh International Conference on Learning Representations. 2022.
2. Zhang Z, Xu M, Lozano A C, et al. Pre-training protein encoder via siamese sequence-structure diffusion trajectory prediction[J]. *Advances in Neural Information Processing Systems*, 2024, 36.
3. Zeng Y, Wei Z, Yuan Q, et al. Identifying B-cell epitopes using AlphaFold2 predicted structures and pretrained language model[J]. *Bioinformatics*, 2023, 39(4): btad187.
4. Kipf T N, Welling M. Semi-Supervised Classification with Graph Convolutional Networks[C]//International Conference on Learning Representations. 2016.
5. Baskaran K, Duarte J M, Biyani N, et al. A PDB-wide, evolution-based assessment of protein-protein interfaces[J]. *BMC Structural Biology*, 2014, 14: 1-11.
6. Duarte J M, Srebniak A, Schärer M A, et al. Protein interface classification by evolutionary analysis[J]. *BMC bioinformatics*, 2012, 13: 1-16.
7. de Jong D H, Periole X, Marrink S J. Dimerization of amino acid side chains: lessons from the comparison of different force fields[J]. *Journal of Chemical Theory and Computation*, 2012, 8(3): 1003-1014.
8. Sengupta D, Marrink S J. Lipid-mediated interactions tune the association of glycoporphin A helix and its disruptive mutants in membranes[J]. *Physical Chemistry Chemical Physics*, 2010, 12(40): 12987-12996.
9. Periole X, Knepp A M, Sakmar T P, et al. Structural determinants of the supramolecular organization of G protein-coupled receptors in bilayers[J]. *Journal of the American Chemical Society*, 2012, 134(26): 10959-10965.
10. Lamprakis C, Andreadelis I, Manchester J, et al. Evaluating the efficiency of the Martini force field to study protein dimerization in aqueous and membrane environments[J]. *Journal of Chemical Theory and Computation*, 2021, 17(5): 3088-3102.
11. Lelimosin M, Limongelli V, Sansom M S P. Conformational changes in the epidermal growth factor receptor: Role of the transmembrane domain investigated by coarse-grained metadynamics free energy calculations[J]. *Journal of the American Chemical Society*, 2016, 138(33): 10611-10622.
12. Roel-Touris J, Don C G, V. Honorato R, et al. Less is more: coarse-grained integrative modeling of large biomolecular assemblies with HADDOCK[J]. *Journal of chemical theory and computation*, 2019, 15(11): 6358-6367.
13. Kroon P C, Grunewald F, Barnoud J, et al. Martinize2 and Vermouth: Unified Framework for Topology Generation[J]. *eLife*, 2023, 12.
14. Monticelli L, Kandasamy S K, Periole X, et al. The MARTINI coarse-grained force field: extension to proteins[J]. *Journal of chemical theory and computation*, 2008, 4(5): 819-834.
15. Burley S K, Berman H M, Kleywegt G J, et al. Protein Data Bank (PDB): the single global

- macromolecular structure archive[J]. *Protein crystallography: methods and protocols*, 2017: 627-641.
16. Liu X, Luo Y, Li P, et al. Deep geometric representations for modeling effects of mutations on protein-protein binding affinity[J]. *PLoS computational biology*, 2021, 17(8): e1009284.
 17. Notin P, Kollasch A, Ritter D, et al. ProteinGym: large-scale benchmarks for protein fitness prediction and design[J]. *Advances in Neural Information Processing Systems*, 2024, 36.
 18. Jumper J, Evans R, Pritzel A, et al. Highly accurate protein structure prediction with AlphaFold[J]. *Nature*, 2021, 596(7873): 583-589.
 19. Evans R, O'Neill M, Pritzel A, et al. Protein complex prediction with AlphaFold-Multimer[J]. *bioRxiv*, 2021: 2021.10.04.463034.
 20. Mirdita M, Schütze K, Moriwaki Y, et al. ColabFold: making protein folding accessible to all[J]. *Nature methods*, 2022, 19(6): 679-682.
 21. Baek M, DiMaio F, Anishchenko I, et al. Accurate prediction of protein structures and interactions using a three-track neural network[J]. *Science*, 2021, 373(6557): 871-876.
 22. Abramson J, Adler J, Dunger J, et al. Accurate structure prediction of biomolecular interactions with AlphaFold 3[J]. *Nature*, 2024: 1-3.
 23. Bryant P, Pozzati G, Elofsson A. Improved prediction of protein-protein interactions using AlphaFold2[J]. *Nature communications*, 2022, 13(1): 1265.
 24. Chen R, Li L, Weng Z. ZDOCK: an initial-stage protein-docking algorithm[J]. *Proteins: Structure, Function, and Bioinformatics*, 2003, 52(1): 80-87.
 25. Yan Y, Zhang D, Zhou P, et al. HDOCK: a web server for protein-protein and protein-DNA/RNA docking based on a hybrid strategy[J]. *Nucleic acids research*, 2017, 45(W1): W365-W373.
 26. Ganea O E, Huang X, Bunne C, et al. Independent SE (3)-Equivariant Models for End-to-End Rigid Protein Docking[C]//International Conference on Learning Representations. 2021.
 27. Singh R, Devkota K, Sledzieski S, et al. Topsy-Turvy: integrating a global view into sequence-based PPI prediction[J]. *Bioinformatics*, 2022, 38(Supplement_1): i264-i272.
 28. Zhang Y, Skolnick J. TM-align: a protein structure alignment algorithm based on the TM-score[J]. *Nucleic acids research*, 2005, 33(7): 2302-2309.

Manuscript ID: NCOMMS-24-15723B

Manuscript Title: Integration of molecular coarse-grained model into geometric representation learning framework for protein-protein complex property prediction

Response to Reviewer 3

We are very grateful for your comments and thoughtful suggestions. We have carefully considered all comments and have revised the manuscript accordingly. All changes made to the text are in blue so that they may be easily identified.

Q1: As pointed out in my initial review, the ability to predict dG is not of great interest, and I seriously doubt that the improvements obtained here will be of any significance when you consider the experimental accuracies. Even trivial methods, such as measuring the buried hydrophobic surface area are likely to do quite well on predicting dG .

Therefore, I would again insist that the author demonstrate that their method can be used to predict $\Delta\Delta G$ values before publishing this paper. If that is not possible that hints at this method not being useful for the scientific community.

Thank you for highlighting important points regarding the prediction of ΔG and $\Delta\Delta G$ values. **ΔG and $\Delta\Delta G$ are inter-related yet distinct metrics in protein-protein interactions (PPIs).** While we fully acknowledge the significance of $\Delta\Delta G$ predictions in assessing the impact of mutations, we also argue that ΔG predictions for PPIs are of great interest. ΔG provides fundamental insights into the stability and binding strength of PPIs, which are crucial for understanding biological processes and drug design. Specifically, predicting ΔG for peptides/antibodies of varying lengths and shapes binding to a target protein allows for rapid screening of candidate peptides/antibodies, a task $\Delta\Delta G$ cannot perform as it mainly predicts the impact of mutations on existing PPIs and cannot evaluate newly and independently designed peptides/antibodies. Additionally, accurate ΔG predictions are essential as they form the basis for reliable $\Delta\Delta G$ predictions. While $\Delta\Delta G$ values are critical for assessing the impact of mutations, they are derived from changes in ΔG ($\Delta\Delta G = \Delta G^{wild-type/WT} - \Delta G^{mutant/MT}$). Improving ΔG predictions directly enhances $\Delta\Delta G$ prediction capabilities.

In response to the Reviewer 3's request, we have conducted an independent experiment to demonstrate the potential of our method MCGLPPI in $\Delta\Delta G$ predictions. We adapted our method into $\Delta\Delta G$ predictions without carefully designed modules and compared its performance with advanced $\Delta\Delta G$ prediction methods.

Dataset: we used a multiple-point mutation dataset AB-bind [1], which contains 1101 sample points related to the binding affinity change (i.e., $\Delta\Delta G$) caused by multiple-point amino acid (AA) mutations

on the complex formed from antibody or antibody-like binding. Following the basic experimental settings of the existing pre-trained graph neural network (GNN)-based approach MpbPPI [2], which had already tested on this dataset, FoldX [3] was chosen to complete all side chains and generate the mutation complex structure based on the corresponding raw PDB file of wild-type (WT) structure and mutational site information (and then all structures were transformed into CG-scale protein graphs based on MARTINI). On top of this, a WT protein-protein complex type-based fivefold cross-validation (CV) (detailed in [2]), which ensures no intersection of original WT protein-protein complex types between any of the two folds, was leveraged for model performance evaluation.

Model Adaption: $\Delta\Delta G$ represents the binding affinity/binding free energy (i.e., ΔG) change from WT to mutant (MT) status (i.e., $\Delta G^{WT} - \Delta G^{MT}$). Inspired by this, keeping other model configurations unchanged, we utilized our pre-trained encoder to process the CG complex graph before mutation and after mutation separately (using the same model parameters). We obtained the graph embedding for both WT (denoted as h^{WT}) and mutation structures (denoted as h^{MT}). After that, the final $\Delta\Delta G$ prediction was estimated based on $h^{WT} - h^{MT}$ followed by the three-layer multi-layer perception (MLP).

Evaluation Metric: We evaluated our method (MCGLPPI-M2), and recorded the prediction results based on the Pearson's correlation coefficient (R_p), the root mean square error (RMSE), and the mean absolute error (MAE). We also reported the corresponding results of MpbPPI and two representative energy-based specialized methods FoldX and Rosetta macromolecular modeling suite (Flex ddG) [4] from the ref [2]. Please see Supplementary Table 5 for the experiment results.

Conclusion: Our method achieved competitive performance compared with advanced $\Delta\Delta G$ prediction methods. It is worth noting that our method only adopts a straightforward adaptation without any specifically designed modules, while the compared methods are carefully proposed for the $\Delta\Delta G$ predictions. This demonstrates the advantages of our method and proves the versatility of our framework in handling various PPI property prediction tasks. We believe that our framework will bring new insights into the scientific community.

We have added this evaluation regarding our method in $\Delta\Delta G$ prediction to the manuscript (see lines 594-596, page 29) and Supplementary Information (see the section of Further extension to protein complex $\Delta\Delta G$ prediction based on simple modifications).

References

1. Sirin S, Apgar J R, Bennett E M, et al. AB-bind: antibody binding mutational database for computational affinity predictions[J]. Protein Science, 2016, 25(2): 393-409.
2. Yue Y, Li S, Wang L, et al. MpbPPI: a multi-task pre-training-based equivariant approach for the prediction of the effect of amino acid mutations on protein-protein interactions[J]. Briefings in Bioinformatics, 2023, 24(5): bbad310.
3. Schymkowitz J, Borg J, Stricher F, et al. The FoldX web server: an online force field[J]. Nucleic

acids research, 2005, 33(suppl_2): W382-W388.

4. Barlow K A, Ó Conchúir S, Thompson S, et al. Flex ddG: Rosetta ensemble-based estimation of changes in protein–protein binding affinity upon mutation[J]. The Journal of Physical Chemistry B, 2018, 122(21): 5389-5399.

Manuscript ID: NCOMMS-24-15723C

Manuscript Title: Integration of molecular coarse-grained model into geometric representation learning framework for protein-protein complex property prediction

Response to Reviewer 3

We are very grateful for your comments and thoughtful suggestions. We have carefully considered all comments and have revised the manuscript accordingly. All changes made to the text are in blue so that they may be easily identified.

Q1: I appreciate the authors' attempts to predict $\Delta\Delta G$ - but in contrast to what they try to claim in this paper, this hints at the low value of the initial predictions. If the initial method actually was useful, there would be no need to use a "three-layer MLP" to predict the $\Delta\Delta G$ from the hWT-hMT. It would just be possible to use $\Delta\Delta G = \Delta G^{WT} - \Delta G^{MT}$ directly without any training. I assume that the authors tried that, but the results were too bad to mention - which is scientifically misleading. Further, it is not clear how homology reduction or cross-validation was performed in this test - it is not clear from Ref 26 if they did use homology reductions on the antigens or not (they should have but from what I can see it is not described in that way).

We thank the reviewer for your comments regarding the necessity of employing a three-layer MLP model, the accuracy of $\Delta\Delta G$ values obtained using $\Delta G^{WT} - \Delta G^{MT}$ directly, and the homology reduction details. Here we address these points as follows:

Comment 1: *"If the initial method actually was useful, there would be no need to use a "three-layer MLP" to predict the $\Delta\Delta G$ from the hWT-hMT. It would just be possible to use $\Delta\Delta G = \Delta G^{WT} - \Delta G^{MT}$ directly without any training. I assume that the authors tried that, but the results were too bad to mention - which is scientifically misleading."*

Reply: We need to clarify the proposed MCGLPPI method is not a predictor for $\Delta\Delta G$. In fact, MCGLPPI is a new general and more efficient protein representation (compared with its atom- and residue-scale counterparts) based on the coarse-grained (CG) protein complex graph originated from the MARTINI force field. Such representation is useful because it can be adapted and adjusted for various downstream protein complex property prediction tasks. In the paper, we demonstrate this by using the "three-layer multi-layer perception (MLP)" as the predictor to decode the corresponding CG representations for ΔG , $\Delta\Delta G$, and interface type predictions.

To address the second part of the comment that “*It would just be possible to use $ddG=dgWT-dgMT$ directly without any training*”, we first need to point out that we adopt the standard task formulation from References [1, 2]. In this task formulation, based on the corresponding protein complex structures in the multiple-point mutation dataset AB-bind [3], the task for the machine learning methods is, during the training phase based on available training samples, the models are directly optimized based on:

- 1) Inputting the pairwise structures or their proxies of the wild-type (WT) and mutant (MT) complexes.
- 2) Outputting the sole $\Delta\Delta G$ values (as the prediction optimization objective) for efficient utilization of the complex representation produced by the protein encoder.

Therefore, there are no explicit WT and MT ΔG labels to guide the model optimization. Related to this, the original AB-bind dataset does not provide the raw WT and MT ΔG labels (<https://onlinelibrary.wiley.com/doi/full/10.1002/pro.2829>), which makes the reviewer’s suggestion “*to use $ddG=dgWT-dgMT$ directly without any training*” impossible in this case. In [1, 2], the binding intensity difference between WT and MT complex structures is implicitly modeled and jointly optimized based on the WT-MT structural representation difference (from the protein encoder). Based on the same training samples, we follow the same aforementioned problem formulation and use “three-layer MLP” to predict the $\Delta\Delta G$ from the representation difference between WT and MT complexes (i.e., $h^{WT} - h^{MT}$) for a fair comparison.

Supplementary Table 5: the $\Delta\Delta G$ prediction results based on the AB-bind dataset

5-fold WT type-based CV	R_p	RMSE	MAE
MCGLPPI	0.451	1.880	1.363
MpbPPI [2]	0.442	1.899	1.357
FoldX [4]	0.273	3.429	2.278
Flex ddG [5]	0.059	4.494	2.875
TM-score splitting	R_p	RMSE	MAE
MCGLPPI	0.415	1.327	1.081
AB-bind 962-subset inference	R_p	RMSE	MAE
MCGLPPI	0.280	2.420	1.563
GearNet-Atom [6]	0.150	2.464	1.606
GearNet-Res [7]	0.074	2.476	1.609
AB-bind complete inference	R_p	RMSE	MAE
MCGLPPI	0.264	2.386	1.511

The bold data signifies the best experimental result under the current comparison group.

Following the same standard task formulation and evaluation settings (detailed below), we can find that MCGLPPI is able to achieve competitive performance compared with the advanced $\Delta\Delta G$ prediction methods. However, to demonstrate our results are scientific sound, we design an additional experiment to implement what you suggested. Specifically, to circumvent the problem of no raw WT and MT ΔG labels in the original dataset, we train MCGLPPI-M2, GearNet-Atom (representative atom-scale method)

[6], and GearNet-Res (representative residue-scale method) [7] ΔG prediction models based on the 915-sample-subset of the independent PDBbind-strict-dimer ΔG dataset described in the main manuscript (see **The binding affinity prediction of formation of strict dimers** section of the manuscript, and prediction is given based on a three-layer MLP to decode the CG representation from the CG protein encoder), and predict $\Delta\Delta G$ for all samples in the AB-bind dataset based on the prediction of ΔG^{WT} minus the prediction of ΔG^{MT} directly without extra training (Supplementary Table 5, please note that for GearNet-Atom and GearNet-Res, 962 out of all test samples can be identified and inferred).

We observe that, although our model is trained with an independent ΔG dataset that does not include specialized WT-MT antigen-antibody pairs with explicit $\Delta\Delta G$ labels to guide the model to directly perceive the subtle binding intensity differences between similarity WT and MT structures, MCGLPPI is still able to give a reasonable accuracy, e.g., competitive to the specifically designed energy-based prediction tool FoldX. It is worth mentioning that our model can be further improved by the training based on the aforementioned “ $h^{WT} - h^{MT}$ ” task formulation using available AB-bind training data with $\Delta\Delta G$ labels. However, we should point out that the exploration and discussion of further direct performance improvement of the existing ΔG model (i.e., trained on the corresponding independent ΔG dataset) for the $\Delta\Delta G$ predictions (such as a separate biologically different $\Delta\Delta G$ dataset) belong to an independent cross-domain optimization problem in the computer science field [8, 9], which is out of the scope of the discussion content of this paper.

Comment 2: “Further, it is not clear how homology reduction or cross-validation was performed in this test - it is not clear from Ref 26 if they did use homology reductions on the antigens or not (they should have but from what I can see it is not described in that way).”

Reply: For the evaluation setting of the aforementioned standard task formulation, following the specially designed $\Delta\Delta G$ prediction method MpbPPI [2], we use the same WT protein-protein complex type-based fivefold cross-validation (CV) introduced in [2] for a fair comparison. This setting reduces the sample similarity between training and test sets by ensuring no intersection of original WT protein-protein complex types (for corresponding WT-MT pairs) between any of the two folds and striving to make the both sample total number and WT protein-protein complex type number assigned for each fold to be as close as possible (detailed in Supplementary Information of ref [2]).

This setting does not include a clear cutoff to quantify the dissimilarity between training and test sets, to further test the stability of our method (based on the same model settings) to respond to what you suggested, we further design an additional experiment based on a data split with the explicit TM-score cutoff. Specifically, we utilize the TM-align alignment tool [10] to calculate a similarity matrix based on

the structural difference between pairwise PDB samples within AB-bind, and extract sample points whose TM-score is lower than 0.5 with other PDB samples as the test set (183 sample points are found in total), and the remaining ones are treated as the training set. In this setting, MCGLPPI still can give a test R_p over 0.4 (Supplementary Table 5).

We have added the aforementioned experimental results, the corresponding discussion, and the evaluation setting details to the manuscript (see lines 589-600, page 29) and Supplementary Information (see the section of **Further extension to protein complex $\Delta\Delta G$ predictions based on simple modifications**).

References

1. Liu X, Luo Y, Li P, et al. Deep geometric representations for modeling effects of mutations on protein-protein binding affinity[J]. PLoS computational biology, 2021, 17(8): e1009284.
2. Yue Y, Li S, Wang L, et al. MpbPPI: a multi-task pre-training-based equivariant approach for the prediction of the effect of amino acid mutations on protein-protein interactions[J]. Briefings in Bioinformatics, 2023, 24(5): bbad310.
3. Sirin S, Apgar J R, Bennett E M, et al. AB-bind: antibody binding mutational database for computational affinity predictions[J]. Protein Science, 2016, 25(2): 393-409.
4. Schymkowitz J, Borg J, Stricher F, et al. The FoldX web server: an online force field[J]. Nucleic acids research, 2005, 33(suppl_2): W382-W388.
5. Barlow K A, Ó Conchúir S, Thompson S, et al. Flex ddG: Rosetta ensemble-based estimation of changes in protein-protein binding affinity upon mutation[J]. The Journal of Physical Chemistry B, 2018, 122(21): 5389-5399.
6. Zhang Z, Xu M, Lozano A C, et al. Pre-training protein encoder via siamese sequence-structure diffusion trajectory prediction[J]. Advances in Neural Information Processing Systems, 2024, 36.
7. Zhang Z, Xu M, Jamasb A, et al. Protein representation learning by geometric structure pretraining[J]. arXiv preprint arXiv:2203.06125, 2022.
8. Zheng Y. Methodologies for cross-domain data fusion: An overview[J]. IEEE transactions on big data, 2015, 1(1): 16-34.
9. Yao H, Yang X, Pan X, et al. Improving Domain Generalization with Domain Relations[C]//The Twelfth International Conference on Learning Representations.
10. Zhang Y, Skolnick J. TM-align: a protein structure alignment algorithm based on the TM-score[J]. Nucleic acids research, 2005, 33(7): 2302-2309.